# Learning richness modulates equality reasoning in neural networks

**William L. Tong & Cengiz Pehlevan**

{`wtong@g`,`cpehlevan@seas`}`.harvard.edu`

School of Engineering and Applied Sciences
Center for Brain Sciences
Kempner Institute for the Study of Artificial and Natural Intelligence
Harvard University, Cambridge, MA 02138

## Abstract

Equality reasoning is ubiquitous and purely abstract: sameness or difference may be evaluated no matter the nature of the underlying objects. As a result, same-different (SD) tasks have been extensively studied as a starting point for understanding abstract reasoning in humans and across animal species. With the rise of neural networks that exhibit striking apparent proficiency for abstractions, equality reasoning in these models has also gained interest. Yet despite extensive study, conclusions about equality reasoning vary widely and with little consensus. To clarify the underlying principles in learning SD tasks, we develop a theory of equality reasoning in multi-layer perceptrons (MLP). Following observations in comparative psychology, we propose a spectrum of behavior that ranges from *conceptual* to *perceptual* outcomes. Conceptual behavior is characterized by task-specific representations, efficient learning, and insensitivity to spurious perceptual details. Perceptual behavior is characterized by strong sensitivity to spurious perceptual details, accompanied by the need for exhaustive training to learn the task. We develop a mathematical theory to show that an MLP's behavior is driven by *learning richness*. Rich-regime MLPs exhibit conceptual behavior, whereas lazy-regime MLPs exhibit perceptual behavior. We validate our theoretical findings in vision SD experiments, showing that rich feature learning promotes success by encouraging hallmarks of conceptual behavior. Overall, our work identifies feature learning richness as a key parameter modulating equality reasoning, and suggests that equality reasoning in humans and animals may similarly depend on learning richness in neural circuits.

**Keywords:** equality reasoning; same-different; neural network; conceptual and perceptual behavior

## 1 Introduction

The ability to reason abstractly is a hallmark of human intelligence. Fluency with abstractions drives both our highest intellectual achievements and many of our daily necessities like telling time, navigating traffic, and planning leisure. At the same time, neural networks have grown tremendously in sophistication and scale. The latest examples exhibit increasingly impressive competency, and the potential to automate the reasoning process itself seems imminent (OpenAI, 2024, 2023; Bubeck et al., 2023; Guo et al., 2025). Nonetheless, it remains unclear to what extent these models are able to reason abstractly, and how consistently they behave (McCoy et al., 2023; Mahowald et al., 2024; Ullman, 2023). To begin answering these questions, we require a principled understanding of how neural networks can reason.

A particularly simple and salient form of abstract reasoning is *equality reasoning*: determining whether two objects are the same or different. The "sense of sameness is the very keel and backbone of our thinking," (James, 1905) promoting its study as a tractable viewport into abstract reasoning across humans and animals (E. A. Wasserman & Young, 2010). Despite many decades of study, the history of equality reasoning abounds with widely varying conclusions. Success at same-different (SD) tasks have been documented in a large number of animals, including non-human primates (Vonk, 2003), honeybees (Giurfa et al., 2001), pigeons (E. A. Wasserman & Young, 2010), crows (Smirnova et al., 2015), and parrots (Obozova et al., 2015). Others, however, have argued that animals employ perceptual shortcuts to solve these tasks like using stimulus variability, and lack a true conception of sameness or difference (Penn et al., 2008). Competence at equality reasoning may require exposure to language or some form of symbolic training (Premack, 1983). Meanwhile, pre-lingual human infants have demonstrated sensitivity to same-different relations (G. F. Marcus et al., 1999; Saffran & Thiessen, 2003; Rabagliati et al., 2019).

Equality reasoning in neural networks is no less debated. G. F. Marcus et al. (1999) discovered that seven-month-old infants succeed at an SD task where neural networks fail, launching a lively debate that continues to present day (Seidenberg et al., 1999; Seidenberg & Elman, 1999; Alhama & Zuidema, 2019). Others have demonstrated severe shortcomings in neural networks directed to solve visual same-different reasoning and relational tasks (Kim et al., 2018; Stabinger et al., 2021; Vaishnav et al., 2022; Webb et al., 2023). Such failures motivate a growing literature in bespoke architectural advancements geared towards relational reasoning (Webb et al., 2023, 2020; Santoro et al., 2017; Battaglia et al., 2018). At the same time, modern large language models routinely solve complex reasoning problems (Bubeck et al., 2023). Their surprising success tempers earlier categorical claims against neural networks' reasoning abilities. Even simple models like multi-layer perceptrons (MLPs) have recently been shown to solve equality and relational reasoning tasks with surprisingly efficacy (A. Geiger et al., 2023; Tong & Pehlevan, 2024).

The lack of consensus on equality reasoning in either organic or silicate brains speaks to the need for a stronger theoretical foundation. To this end, we present a theory of equality

reasoning in MLPs that highlights the central role of a hitherto overlooked parameter: *learning richness*, a measure of how much internal representations change over the course of training (Chizat et al., 2019). We find that MLPs in a rich learning regime exhibit *conceptual behavior*, where they develop salient, *conceptual* representations of sameness and difference, learn the task from few training examples, and remain largely insensitive to spurious perceptual details. In contrast, lazy regime MLPs exhibit *perceptual behavior*, where they solve the task only after exhaustive training and show strong sensitivity to perceptual variations. Our specific contributions are the following.

**Contributions**

- We hand-craft a solution to our same-different task that is expressible by an MLP, demonstrating the possibility for our model to solve this task. Our solution suggests what conceptual representations may look like, guiding subsequent analysis.

- We argue that an MLP trained in a rich feature learning regime attains the hand-crafted solution, and exhibits three hallmarks of conceptual behavior: conceptual representations, efficient learning, and insensitivity to spurious perceptual details.

- We prove that an MLP trained in a lazy learning regime can also solve an equality reasoning task, but exhibits perceptual behavior: it requires exhaustive training data and shows strong sensitivity to spurious perceptual details.

- We extend our results to same-different tasks with noise, calculating Bayes optimal performance under priors that either generalize to arbitrary inputs or memorize the training set. We demonstrate that rich MLPs attain Bayes optimal performance under the generalizing prior.

- We validate our results on complex visual SD tasks, showing that our theoretical predictions continue to hold.

Our theory clarifies the understudied role of learning richness in driving successful reasoning, with potential implications for both neural network design and animal cognition.

## 1.1 Related work

In studying same-different tasks comparatively across animal species, E. Wasserman et al. (2017) observe a continuum between perceptual and conceptual behavior. Some animals focus on spurious perceptual details in the task stimuli like image variability, and slowly gain competence through exhaustive repetition. Other animals and humans appear to develop a conceptual understanding of sameness, allowing them to learn the task quickly and ignore irrelevant percepts. Many others fall somewhere in between, exhibiting behavior with both perceptual and conceptual components. These observations lend themselves to a theory where representations

and learning mechanisms operate over a continuous domain (Carstensen & Frank, 2021).

Neural networks offer a natural instantiation of such a continuous theory. However, the extent to which neural networks can reason at all remains a hotly contested topic. Famously, Fodor & Pylyshyn (1988) argue that connectionist models are poorly equipped to describe human reasoning. G. F. Marcus (1998) further contends that neural networks are altogether incapable of solving many simple symbolic tasks (see also G. F. Marcus et al. (1999); G. F. Marcus (2003); G. Marcus (2020)). Boix-Adsera et al. (2023) have also argued that MLPs are unable to generalize on relational problems like our same-different task, though this finding has been contested (Tong & Pehlevan, 2024; A. Geiger et al., 2023).

Negative assertions about neural network reasoning appear to weaken when considering modern LLMs, which routinely solve complex math and logic problems (Bubeck et al., 2023; OpenAI, 2024; Guo et al., 2025). But even here, doubts remain about whether LLMs truly reason or merely reproduce superficial aspects of their enormous training set (McCoy et al., 2023; Mahowald et al., 2024). Nonetheless, A. Geiger et al. (2023) found that simple MLPs convincingly solve same-different tasks after moderate training. Tong & Pehlevan (2024) further showed that MLPs solve a wide variety of relational reasoning tasks. We support these findings by arguing that MLPs solve a same-different task, but their performance is modulated by learning richness. In resonance with E. Wasserman et al. (2017), varying richness pushes an MLP along a spectrum between a perceptual and conceptual solutions to the task.

Learning richness itself refers to the degree of change in a neural network's internal representations during training. A number of network parameters were recently discovered to control learning richness, including the readout scale, initialization scheme, and learning rate (Chizat et al., 2019; Woodworth et al., 2020; Yang & Hu, 2021; Bordelon & Pehlevan, 2022). In the brain, learning richness may correspond to forming adaptive representations that encode task-specific variables, in contrast to fixed representations that remain task agnostic (Farrell et al., 2023). Studies have used learning richness to understand the neural representations underlying diverse phenomena like context-dependent decision making (Flesch et al., 2022), multitask cognition (Ito & Murray, 2023), generalizing knowledge to new tasks (Johnston & Fusi, 2023), and even consciousness (Mastrovito et al., 2024).

## 2 Setup

We consider the following same-different task, inspired by the setup in A. Geiger et al. (2023). The task consists of input pairs $\mathbf{z}_1, \mathbf{z}_2 \in \mathbb{R}^d$, where $\mathbf{z}_i = \mathbf{s}_i + \eta_i$. The labeling function $y$ is given by

$$y(\mathbf{z}_1, \mathbf{z}_2) = \begin{cases} 1 & \mathbf{s}_1 = \mathbf{s}_2 \\ 0 & \mathbf{s}_1 \neq \mathbf{s}_2 \end{cases}.$$

Quantities $\mathbf{s}$ correspond to "symbols," perturbed by a small amount of noise $\eta$. Noise is distributed as $\eta \sim \mathcal{N}(\mathbf{0}, \sigma^2 \mathbf{I}/d)$,

for some choice of $\sigma^2$. Initially we will take $\sigma^2 = 0$ so $\mathbf{z} = \mathbf{s}$, but we will allow $\sigma^2$ to be nonzero when considering a noisy extension to the task. Our definition of equality implies exact identity, up to possible noise. Other commonly studied variants include equality up to transformation (Fleuret et al., 2011), hierarchical equality (Premack, 1983), context-dependent equality (Raven, 2003), among many others. We pursue exact identity for its tractability and ubiquity in the literature, and investigate more general notions of equality later with experiments in the noisy case and in vision tasks.

The model consists of a two-layer MLP without bias parameters

$$f(\mathbf{x}) = \frac{1}{\gamma\sqrt{d}} \sum_{i=1}^{m} a_i \phi(\mathbf{w}_i \cdot \mathbf{x}), \qquad (1)$$

where $\phi$ is a ReLU activation applied point-wise to its inputs. We use the standard logit link function to produce predictions $\hat{y} = 1/(1 + e^{-f})$. Inputs are concatenated as $\mathbf{x} = (\mathbf{z}_1; \mathbf{z}_2) \in \mathbb{R}^{2d}$ before being passed to $f$. The model is trained using binary cross entropy loss with a learning rate $\alpha = \gamma^2 d \alpha_0$, for a fixed $\alpha_0$. Hidden weight vectors are initialized as $\mathbf{w}_i \sim \mathcal{N}(\mathbf{0}, \mathbf{I}/m)$, and readouts as $a_i \sim \mathcal{N}(0, 1/m)$. To enable interpolation between rich and lazy learning regimes, the MLP is centered such that $f(\mathbf{x}) = 0$ at initialization, for all inputs $\mathbf{x}$. We use a standard procedure for centering, described in Appendix F. Occasionally, we gather all readouts $a$ and hidden weights $\mathbf{w}$ into a single set $\theta$, and write $f(\mathbf{x}; \theta)$ to mean an MLP $f$ parameterized by weights $\theta$. We avoid considering bias parameters to simplify the analysis. In practice, because the task is symmetric about the origin, we find that bias plays little role.

The parameter $\gamma$ controls *learning richness*, where higher values of $\gamma$ correspond to greater richness (Chizat et al., 2019; M. Geiger et al., 2020; Woodworth et al., 2020; Bordelon & Pehlevan, 2022). A neural network trained in a *rich regime* experiences significant changes to its hidden activations $\phi(\mathbf{w_i} \cdot \mathbf{x})$, resulting in task-specific representations. In contrast, a neural network trained in a *lazy regime* retains task-agnostic representations determined by their initialization. The limit $\gamma \to 0$ induces lazy behavior. Increasing $\gamma$ increases learning richness. For our tasks, we find that $\gamma = 1$ produces sufficiently rich learning[1], and increasing $\gamma$ beyond 1 does not qualitatively change our results (Figure C3). Appendix F elaborates on our scaling scheme.

Crucially, the training set consists of a finite number of symbols $\mathbf{s}_1, \mathbf{s}_2, \ldots, \mathbf{s}_L$. These $L$ symbols are sampled before training begins as $\mathbf{s} \sim \mathcal{N}(\mathbf{0}, \mathbf{I}/d)$, then used exclusively to train the model. Training examples are balanced such that half consist of *same* examples and half consist of *different* examples. During testing, symbols $\mathbf{s}$ are sampled afresh for every input,

---

[1]$\gamma = 1$ produces a scaling that is similar to $\mu$P or mean-field parametrization common elsewhere in the rich learning literature (Yang & Hu, 2021; Mei et al., 2018; Rotskoff & Vanden-Eijnden, 2022). However, these scalings technically consider an infinite *width* limit. Our setting considers an infinite *input dimension* limit (Biehl & Schwarze, 1995; Saad & Solla, 1995; Goldt et al., 2019), resulting in an extra $1/\sqrt{d}$ prefactor that is not present in these other scalings.

measuring the model's ability to generalize on *unseen* test examples. If a model has learned equality reasoning, then it should attain perfect test accuracy despite having never witnessed the particular inputs. When $\sigma^2 = 0$, this procedure is precisely equivalent to using one-hot encoded symbol inputs with a fixed embedding matrix, where the model is trained on a subset of all possible symbols. Additional details on our model and setup are enumerated in Appendix G.

## 2.1 Conceptual and perceptual behavior

Central to our framework is the distinction between conceptual and perceptual behavior. Conceptual behavior refers to a facility with abstract concepts, enabling the reasoner to learn an abstract task quickly and generalize with limited dependency on spurious details. Perceptual behavior refers to the opposite, where the reasoner solves a task through sensory association. Such learning is typically characterized by exhaustive training and marked sensitivity to spurious perceptual details.

We posit that learning richness moves an MLP between conceptual and perceptual behavior. We identify three specific characteristics of a conceptual outcome:

1. **Conceptual representations.** We look for evidence of task-specific representations that denote sameness or difference. Such representations should be crucial to solving the task, and contribute towards the model's efficiency and insensitivity to spurious perceptual details (below).

2. **Efficiency.** We measure learning efficiency using the number of different symbols $L$ observed during training. A conceptual reasoner should solve the task with a smaller $L$ than a perceptual reasoner.

3. **Insensitivity to spurious perceptual details.** Spurious perceptual details refer to aspects of the task that influence the input but not the correct output. A readily measurable example is the input dimension $d$. Sameness or difference can be evaluated regardless of $d$. A conceptual reasoner should perform equally well when training on tasks across a variety of $d$, whereas a perceptual reasoner may find certain $d$ harder to learn with than others. We therefore evaluate this insensitivity by comparing the test accuracy of models trained across a large range of input dimensions.

A perceptual solution is characterized by the negation of each point: it does not develop task-specific representations, it requires a large $L$ to solve the task, and test accuracy changes substantially with $d$. While potentially possible to have a mixed solution that exhibits a subset of these points, we do not observe them in practice, and the conceptual/perceptual distinction is sufficiently descriptive of our model.

## 3 Same-different task analysis

We present our analysis of the SD task. We first hand-craft a solution that is expressible by our MLP, and in the process suggest what conceptual representations of sameness and difference may look like (Section 3.1). We proceed to argue that

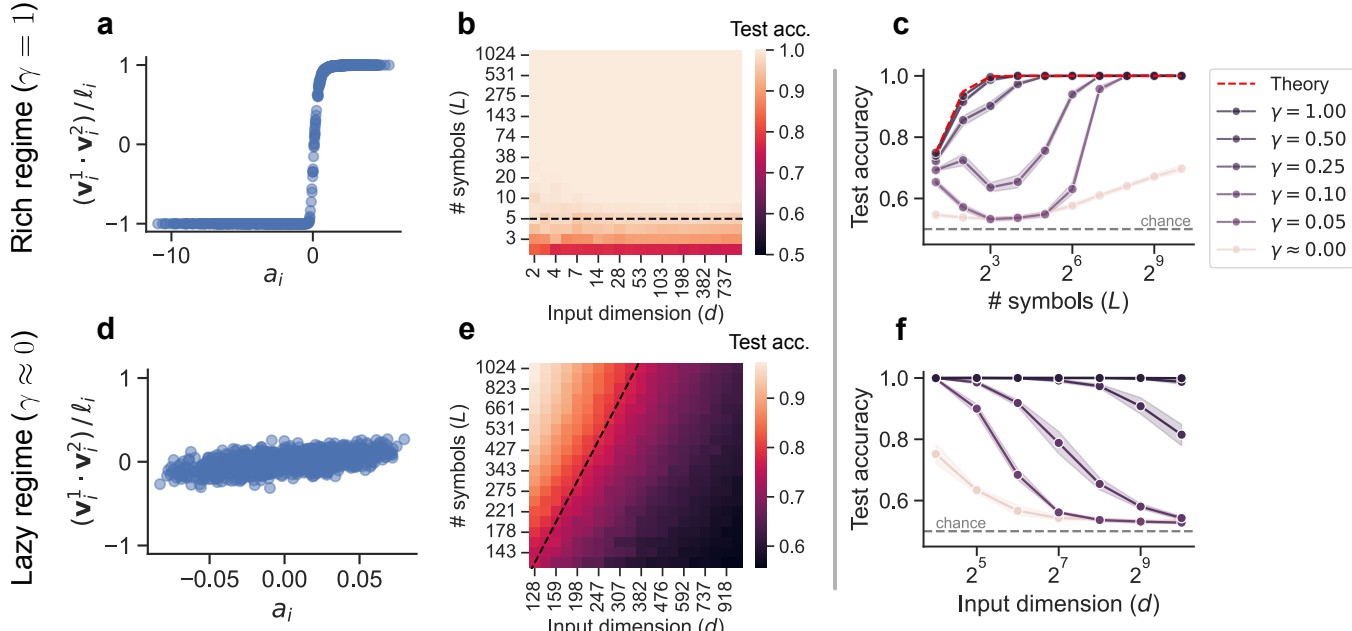

Figure 1: **Rich and lazy regime simulations.** We confirm our theoretical predictions with numeric simulations. **(a).** Hidden weight alignment plotted against readout weights for a rich model. Weights become parallel or antiparallel, with generally higher magnitudes among negative readouts. **(b)** Test accuracy across different input dimensions and training symbols for a rich model ($m = 4096$). Accuracy is not affected by input dimension. **(c)** Test accuracy across different numbers of training symbols, for varying learning richness ($d = 256, m = 1024$). Richer models attain high performance with substantially fewer training symbols. The theoretically predicted rich test accuracy shows excellent agreement with our richest model. Finer-grain validation is plotted in Figure C3. **(d)** Hidden weight alignment plotted against readout weights for a lazy model. There is some correlation between alignment and readout weight, but the weights are nowhere near as close to being parallel/antiparallel as in the rich regime. **(e)** Test accuracy across input dimensions and training symbols for a lazy model ($m = 4096$). Accuracy is substantially affected by input dimension. Theory predicts that the number of training symbols required to maintain high accuracy scales at worst as $L \propto d^2$, plotted in black. **(f)** Test accuracy across different input dimensions, for varying learning richness ($L = 16, m = 1024$). Richer models show less performance decay with increasing dimension. **(all)** Results are computed across six runs. Shading corresponds to empirical 95 percent confidence intervals.

a rich-regime MLP attains the hand-crafted solution through training. It leverages its conceptual representations to learn the task with few training symbols and insensitivity to the input dimension (Section 3.2), exhibiting conceptual behavior. In contrast, a lazy-regime MLP is unable to adapt its representations to the task, and consequently incurs a high training cost and substantial sensitivity to input dimension (Section 3.3), exhibiting perceptual behavior. In an extension to a noisy version of our task, we show that a rich MLP approaches Bayes optimal performance under a generalizing prior across different noise variance $\sigma^2$ (Section 3.3). We validate our results on more complex, image-based tasks in Section 4, and discuss broader implications in Section 5.

## 3.1 Hand-crafted solution

To establish whether an MLP can solve the same-different task at all, we first outline a hand-crafted solution using $m = 4$ hidden units. Let $\mathbf{1} = (1, 1, \ldots, 1) \in \mathbb{R}^d$. Define the weight vector $\mathbf{w}_1^+$ by concatenation: $\mathbf{w}_1^+ = (\mathbf{1}; \mathbf{1}) \in \mathbb{R}^{2d}$. Further define $\mathbf{w}_2^+ = (-\mathbf{1}; -\mathbf{1})$, $\mathbf{w}_1^- = (\mathbf{1}; -\mathbf{1})$, and $\mathbf{w}_2^- = (-\mathbf{1}; \mathbf{1})$. Let

$a^+ = 1$ and $a^- = \rho$, for some value $\rho > 0$. Our MLP is given by

$$f(\mathbf{x}) = a^+ \left( \phi(\mathbf{w}_1^+ \cdot \mathbf{x}) + \phi(\mathbf{w}_2^+ \cdot \mathbf{x}) \right)$$
$$- a^- \left( \phi(\mathbf{w}_1^- \cdot \mathbf{x}) + \phi(\mathbf{w}_2^- \cdot \mathbf{x}) \right). \quad (2)$$

Note that the weight vectors $\mathbf{w}_1^+, \mathbf{w}_2^+$, which correspond to the positive readout $a^+$, are *parallel*: their components point in the same direction with the same magnitude. Meanwhile, weight vectors $\mathbf{w}_1^-, \mathbf{w}_2^-$ corresponding to the negative readout $a^-$ are *antiparallel*: their components point in exact opposite directions with the same magnitude. Only the sign of $f$ impacts the classification, so we assign $a^+ = 1$ and $a^- = \rho$ to represent the relative magnitude of $a^-$ against $a^+$.

To see how this weight configuration solves the same-different task, suppose we receive a *same* example $\mathbf{x} = (\mathbf{z}, \mathbf{z})$. Plugging this into Eq (2) reveals that the negative terms vanish through our antiparallel weights, leaving $f(\mathbf{x}) = 2|\mathbf{1} \cdot \mathbf{z}| > 0$, correctly classifying this example.

Now suppose we receive a *different* example $\mathbf{x}' = (\mathbf{z}, \mathbf{z}')$. Recall that these quantities are sampled independently as

$\mathbf{z}, \mathbf{z}' \sim \mathcal{N}(\mathbf{0}, \mathbf{I}/d)$. As a result, we can no longer rely on a convenient cancellation. The quantity $\phi(\mathbf{w}_1^+ \cdot \mathbf{x}') + \phi(\mathbf{w}_2^+ \cdot \mathbf{x}')$ is equal in distribution to the quantity $\phi(\mathbf{w}_1^- \cdot \mathbf{x}') + \phi(\mathbf{w}_2^- \cdot \mathbf{x}')$, with respect to the randomness in $\mathbf{x}'$. Hence, to implement a consistent negative classification, we need to raise the relative magnitude $\rho$ of our negative readout weight. Indeed, we calculate $p(f(\mathbf{x}') < 0) = \frac{2}{\pi}\tan^{-1}(\rho)$, which approaches 1 for $\rho \gg 1$.[2] Hence, by maintaining a large negative readout, we classify negative examples correctly with high probability. An illustration of this solution is provided in Figure B1.

An MLP need not implement this precise weight configuration to solve the SD task. Rather, our hand-crafted solution suggests two general conditions:

1. **Parallel/antiparallel weight vectors**. Weights associated with positive readouts must be parallel, and weight associated with negative readouts must be antiparallel. This allows us to classify any *same* example by canceling the contribution from negative readouts.

2. **Large negative readouts**. The cumulative magnitude of the negative readouts must be larger than that of the positive readouts. This allows us to classify any *different* example by raising the contribution from negative readouts.

Observe also that parallel and antiparallel weights are suggestive of conceptual representations for sameness and difference. Parallel weights contribute to a *same* classification, and exemplify the structure of a *same* example: the two components point in the same direction. Antiparallel weights contribute to a *different* classification, and exemplify the structure of a *different* example: the two components point as far apart as possible. We look for parallel/antiparallel weight vectors as evidence for conceptual representations of our SD task.

## 3.2 Rich regime

The rich learning regime is characterized by substantial weight changes throughout the course of training. For the MLP given in Eq (1), larger values of $\gamma$ lead to rich learning behavior. We allow $\gamma$ to vary between 0 and 1. The range $\gamma > 1$ is considered in Figure C3, where we see that no qualitative changes to our results occur for larger values of $\gamma$.

To study the rich regime, we take two approaches. First, recent theoretical work (Morwani et al., 2023; Wei et al., 2019; Chizat & Bach, 2020) suggest that MLPs trained in a rich learning regime on a classification task discover a max margin solution: the weights maximize the distance between training points of different classes. We derive the max margin weights for an MLP with quadratic activations in Theorem 1, finding that the max margin solution consists of parallel/antiparallel weight vectors, just as required from our hand-crafted solution. We defer the proof of this theorem to Appendix C.

**Theorem 1.** *Let $\mathcal{D} = \{\mathbf{x}_n, y_n\}_{n=1}^P$ be a training set consisting of $P$ points sampled across $L$ training symbols, as specified in Section 2. Let $f$ be the MLP given by Eq 1, with two changes:*

1. *Fix the readouts $a_i = \pm 1$, where exactly $m/2$ readouts are positive and the remaining are negative.*

2. *Use quadratic activations $\phi(\cdot) = (\cdot)^2$.*

*For weights $\theta = \{\mathbf{w}_i\}_{i=1}^m$, define the max margin set $\Delta(\theta)$ to be*

$$\Delta(\theta) = \arg\max_{\theta} \frac{1}{P} \sum_{n=1}^P \left[ (2y_n - 1) f(\mathbf{x}_n; \theta) \right],$$

*subject to the norm constraints $||\mathbf{w}_i|| = 1$. If $P, L \to \infty$, then for any $\mathbf{w}_i = (\mathbf{v}_i^1; \mathbf{v}_i^2) \in \Delta(\theta)$ and $\ell_i = ||\mathbf{v}_i^1|| \, ||\mathbf{v}_i^2||$, we have that $\mathbf{v}_i^1 \cdot \mathbf{v}_i^2/\ell_i = 1$ if $a_i = 1$ and $\mathbf{v}_i^1 \cdot \mathbf{v}_i^2/\ell_i = -1$ if $a_i = -1$. Further, $||\mathbf{v}_i^1|| = ||\mathbf{v}_i^2||$.*

However, the max margin result does not use ReLU MLPs, relies on fixed readouts $a_i$, and says nothing about learning efficiency or insensitivity to spurious perceptual details, two additional properties we require from a conceptual solution. To address these shortcomings, we extend the analysis by proposing a heuristic construction that approximates a rich ReLU MLP as an ensemble of independent Markov processes (Section C.2). Doing so enables a deeper characterization of rich learning dynamics, resulting in the following approximation of the test accuracy. Given an unseen test point $\mathbf{x}, y$, and prediction $\hat{y}$,

$$p(y = \hat{y}(\mathbf{x})) \approx \frac{1}{2} + \frac{1}{2}\Phi\left( \sqrt{\frac{2(L^2 - L)}{13(\pi - 2)}} \right), \tag{3}$$

where $L$ is the number of training symbols and $\Phi$ is the CDF of a standard normal distribution.[3] This estimate suggests that the model attains over 95 percent test accuracy with as few as $L = 5$ training symbols, and test accuracy does not change with different $d$.

We confirm our theoretical predictions with simulations in Figure 1. At the end of training, the hidden weights indeed become parallel and antiparallel, with negative coefficients gaining larger magnitude (Figure 1a). Figures 1b and c show that the rich model learns the same-different task with substantially fewer training symbols than lazier models, and exhibits excellent agreement with our theoretical test accuracy prediction. As predicted, the rich model's performance does not vary with input dimension (Figure 1b).

Altogether, the rich model develops conceptual representations, learns the same-different task given only a small number of training symbols, and exhibits clear insensitivity to input dimension. In this way, it exhibits conceptual behavior on the same-different task.

## 3.3 Lazy regime

The lazy learning regime is characterized by vanishingly small change in the model's hidden representations after training. Smaller values of $\gamma$ lead to lazy learning behavior. The limit $\gamma \to 0$ corresponds to the Neural Tangent Kernel (NTK)

---

[2]Full details are recorded in Appendix B.

[3]This estimate is for $L \geq 3$. For $L = 2$, $p(y = \hat{y}) = 3/4$. See Section C.5 for details.

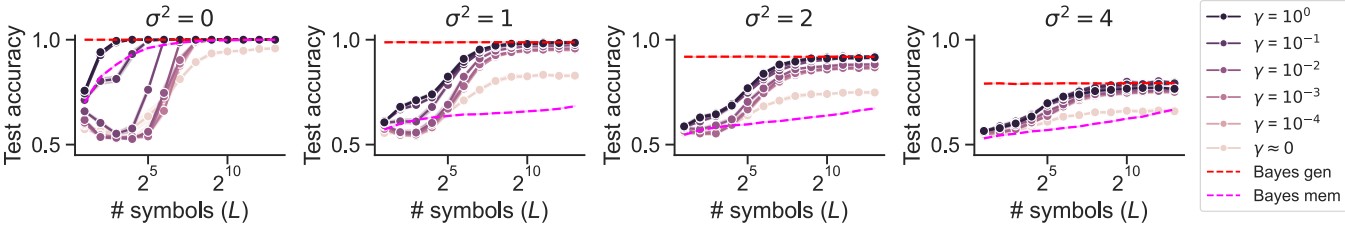

Figure 2: **Bayesian simulations.** Test accuracy across different numbers of training symbols, for varying richness and noise ($d = 64$, $m = 1024$). Bayes optimal accuracy for both generalizing and memorizing priors are plotted with dashed lines. In all cases, rich models attain the Bayes optimal test accuracy under a generalizing prior after sufficiently many training symbols. Shaded error regions are computed across six runs and correspond to empirical 95 percent confidence intervals.

regime, where the network is well-described by a linearization around its initialization (Jacot et al., 2018). In our numerics, we approximate this limit by using $\gamma = (1 \times 10^{-5})/\sqrt{d}$.

Because a lazy neural network cannot adapt its representations to an arbitrary pattern, it is impossible for a lazy MLP to learn parallel/antiparallel weights. However, because the statistics of a *same* example differ from that of a *different* example, it may still be possible for a lazy MLP to succeed at the task given enough training data[4]. Using standard kernel arguments (Cho & Saul, 2009; Jacot et al., 2018), we bound the test error of a lazy MLP in Theorem 2. The proof is deferred to Appendix D.

**Theorem 2** (informal). *Let $f$ be an infinite-width ReLU MLP. If $f$ is trained on a dataset consisting of $P$ points constructed from $L$ symbols with input dimension $d$, then the test error of $f$ is upper bounded by $O\left(\exp\left\{-L/d^2\right\}\right)$.*

This bound suggests that to maintain a consistently low test error (or equivalently, high test accuracy), the number of training symbols $L$ needs to scale quadratically (at worst) with the input dimension: $L \propto d^2$.

We support our theoretical predictions with simulations in Figure 1. Because the model is in a lazy regime, the hidden weights do not move far from initialization, and no clear parallel/antiparallel structure emerges (Figure 1d). Figure 1c shows how models require increasingly more training data as richness decreases. Lazier models are also substantially more impacted by changes in input dimension (Figure 1e and f), and the scaling of training symbols with input dimension is consistent with our theory (Figure 1e).

Altogether, the lazy model is unable to learn conceptual representations, instead relying on statistical associations that require a large amount of training data to learn and exhibit strong sensitivity to input dimension. In this way, the lazy model exhibits perceptual behavior on the same-different task.

### 3.4 Same-different with noise

Up until now, we defined equality by exact identity: even a minuscule deviation in a single coordinate is enough to break

equality and classify an example as *different*. Reality is far less clean, and real-world objects are rarely equal up to exact identity. As a first step towards this broader setting, we relax our dependence on exact identity and consider a noisy SD task. In the notation of our setup (Section 2), we allow $\sigma^2 > 0$.

To understand optimal performance under noise, we apply the following Bayesian framework. As a baseline, we consider a prior corresponding to an idealized model which *memorizes* the training symbols. This memorizing prior assumes every input symbol is distributed uniformly among the training symbols. To contrast this baseline, we consider a gold-standard prior corresponding to a model which *generalizes* to novel symbols. This generalizing prior assumes every input symbol follows the true underlying distribution. By comparing the test accuracy of the trained models to the posteriors computed in these two settings, we identify which prior more closely reflects the models' operation. The calculation of these posteriors are recorded in Appendix E.

Results are plotted in Figure 2. In all cases, we find that the rich model approaches Bayes optimal under the generalizing prior. Lazier models tend to plateau at lower test accuracies; they nonetheless tend to exceed the performance of the memorizing prior at higher noise, indicating some level of generalization. Overall, learning richness appears to support convincing generalization to novel training symbols in the noisy SD task.

## 4 Validation in vision tasks

To validate our theoretical findings in a more complex, naturalistic setting, we turn to visual same-different tasks. Specifically, we examine three datasets designed originally to study visual reasoning and computer vision: 1) PSVRT (Kim et al., 2018), 2) Pentomino (Gülçehre & Bengio, 2016), and 3) CIFAR-100 (Krizhevsky & Hinton, 2009). These tasks offer significantly more challenge over the simple SD task we examine before. Rather than reason over symbol embeddings, a model must now reason over complex visual objects. Inputs are now images, and equality is no longer exact identity: inputs can be equal up to translation (in PSVRT), rotation (in Pentomino), or merely share a class label (CIFAR-100). All additional details on model and task configurations are enumerated in Appendix G.

---

[4]For example, for a *same* input $\mathbf{x} = (\mathbf{z}; \mathbf{z})$ and a *different* input $\mathbf{x}' = (\mathbf{z}_1, \mathbf{z}_2)$, the variance of $\mathbf{1} \cdot \mathbf{x}$ is twice that of $\mathbf{1} \cdot \mathbf{x}'$. Leveraging distinct statistics like this may still allow the lazy model to learn this task.

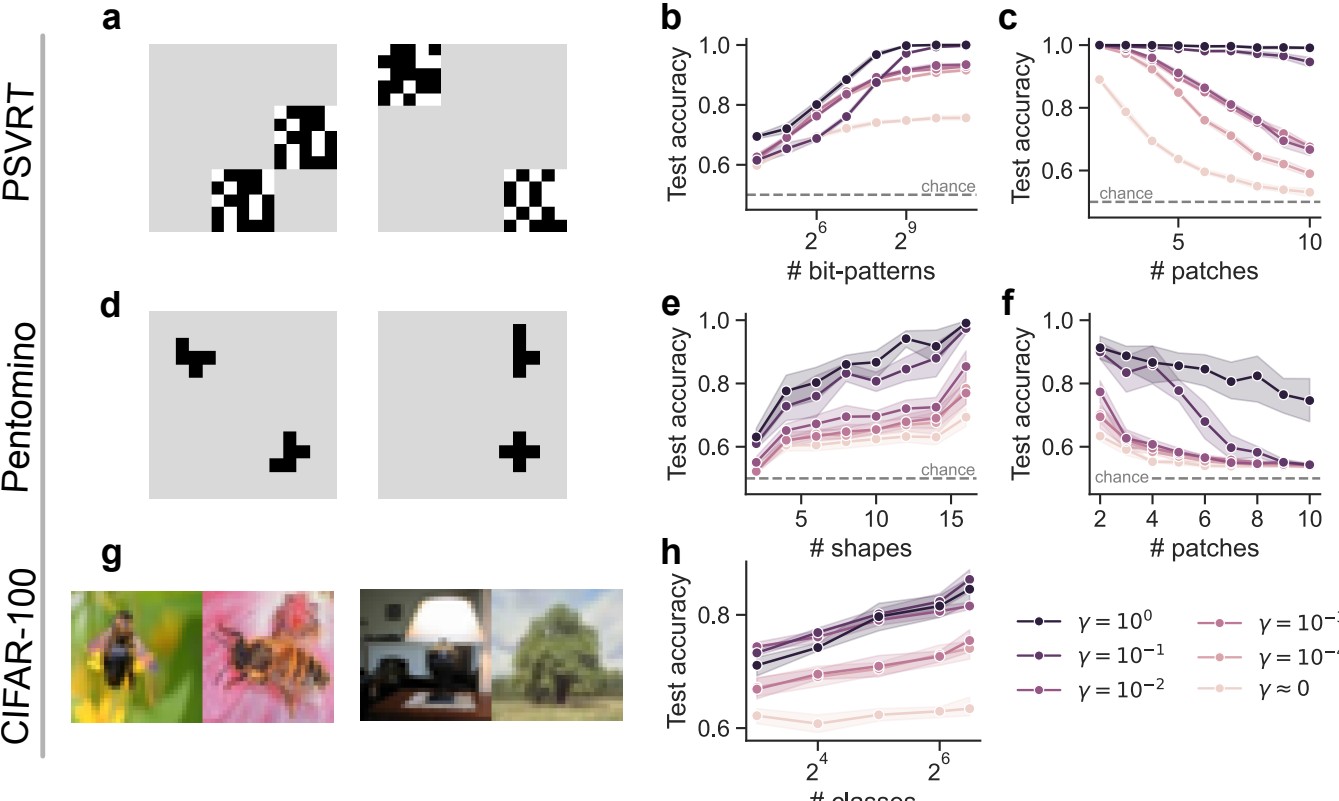

Figure 3: **Visual same-different results. (a)** PSVRT examples for *same* (left) and *different* (right). **(b,c)** Test accuracy on PSVRT across different numbers of training bit-patterns and image widths. Richer models learn the task with fewer patterns and exhibit less sensitivity to larger sizes. **(d)** Pentomino examples for *same* (left) and *different* (right). **(e,f)** Test accuracy on Pentomino across training shapes and image widths. As before, richer models learn the task with fewer training shapes and exhibit less sensitivity to larger sizes, though performance across models tends to diminish somewhat with increasing image size. **(g)** CIFAR-100 examples for *same* (left) and *different* (right). **(h)** Test accuracy on CIFAR-100 same-different across training classes. Richer models tend to perform better with fewer classes, though the richest model in this example performs worse. For this task, very rich models may overfit, necessitating an optimal richness level. **(all)** Shaded error regions are computed across six runs and correspond to empirical 95 percent confidence intervals.

We continue to use the same MLP model as before. Images are flattened before input to the model. Though better performance may be attained using CNNs or Vision Transformers, our ultimate goal is to study learning richness rather than maximize performance.[5] To validate our theoretical findings, we should continue to see the three hallmarks of a conceptual solution (conceptual representations, efficiency, and insensitivity to spurious perceptual details), but only in rich MLPs.

### 4.1 PSVRT

The parameterized-SVRT (PSVRT) dataset is a version of the Synthetic Visual Reasoning Test (SVRT), a collection of challenging visual reasoning tasks based on abstract shapes (Kim et al., 2018). PSVRT replaces the original shapes with random bit-patterns in order to better control image variability. The task input consists of an image that has two blocks of bit-

---
[5]Nonetheless, as we will soon see, an MLP performs astonishingly well on these tasks despite its simplicity — provided it remains in a rich learning regime.

patterns, placed randomly on a blank background. The model must determine whether the blocks contain the same bit-pattern, or different patterns. The training set consists of a fixed number of predetermined bit patterns. The test set consists of novel bit-patterns never encountered during training.

Bit-patterns are *patch-aligned*: they occur in non-overlapping locations that tile the image. The width of an image may be specified by the number of patches. Figure 3a illustrates examples from PSVRT that are three patches wide.

**Results.** Figure 3b plots a model's test accuracy on PSVRT as a function of the number of training patterns. As our theory suggests, richer models learn the task more easily and generalize after substantially fewer training patterns. To test our models under perceptual variation, we consider larger image sizes. We keep the same size of bit-patterns, but increase the number of patches to make a bigger input. Figure 3c indicates that a rich model continues to perform perfectly irrespective of image size, whereas lazier models exhibit a performance de-

cay with larger inputs.

Finally, we identify parallel/antiparallel analogs for PSVRT in the weights of a rich model (Figure A1a). The presence of these conceptual representations suggests that our theory remains a reasonable description for how a rich MLP may learn a conceptual solution to the PSVRT same-different task.

## 4.2 Pentomino

The Pentomino task uses inputs that are *pentomino polygons*: shapes consisting of five squares glued by edge (Gülçehre & Bengio, 2016). The input consists of an image with two pentominoes, placed arbitrarily on a blank background. The pentominoes may either be the same shape, or different. In contrast with the PSVRT task, sameness in this task implies equality up to rotation. After training on a fixed set of pentomino shapes, the model must generalize to entirely novel shapes. Like with PSVRT, shapes are patch-aligned. Figure 3d illustrates example inputs from this task that are three patches wide.

**Results.** Figure 3e plots a model's test accuracy on Pentomino as a function of the number of training shapes. Consistent with our theory, richer models learn the task more easily and generalize after substantially fewer training shapes. To test our models under perceptual variation, we consider larger image sizes. Like with PSVRT, we add additional patches to enlarge the input. Figure 3f indicates that a rich model continues to perform well on larger image sizes, though its performance does start to decay somewhat. Performance decays substantially faster for lazier models.

We again identify parallel/antiparallel analogs for Pentomino in the weights of a rich model (Figure A1c). The presence of these conceptual representations continues to support our theoretical perspective. Notably, Gülçehre & Bengio (2016) introduced this task to motivate curriculum learning, finding that their MLP fails to perform above chance. We found that curriculum learning is unnecessary in the presence of sufficient richness.

## 4.3 CIFAR-100

The CIFAR-100 dataset consists of 60 thousand real-world images, each 32 by 32 pixels (Krizhevsky & Hinton, 2009). Images belong to one of 100 different classes. In this task, the input consists of two different unlabeled images that belong either to the same or different classes. After training on images from a fixed set of labels, the model must generalize to entirely novel labels. The sets of train and test labels are disjoint, making this an extremely challenging task. The labels themselves are not provided in any form during training. Example inputs are illustrated in Figure 3g. We also experiment with providing features from VGG-16 pretrained on ImageNet (Simonyan & Zisserman, 2014). We pass CIFAR-100 images to VGG-16, then use intermediate features as inputs to our MLP. The weights of VGG-16 are fixed throughout the whole process. Note also that ImageNet is disjoint from CIFAR-100, so there is limited possibility of contamination in the test images.

**Results.** Figure 3h plots a model's test accuracy on CIFAR-100 images as a function of the number of training classes. We use outputs from VGG-16 block 4, layer 3, which performed the best with our model. As before, richer models tend to perform better with fewer training classes, but with a curious exception: in contrast to the previous two tasks, the richest model does not always perform decisively the best. This is particularly evident using the activations from other intermediate VGG layers, plotted in Figure G1. For certain layers and number of training classes, the optimal $\gamma$ appears to be somewhat less than 1. This outcome may be in part an artifact of overfitting. Given the complexity of the task and the limited data, richer models are plausibly more susceptible to idiosyncratic features of the training set that generalize poorly, analogous to overfitting effects in classical statistics that degrade the performance of powerful models. In this case, slightly less learning richness may be the optimal setting. Since CIFAR-100 images are fixed to 32 by 32 pixels, we skipped testing variable image size for this task.

As before, we identify parallel/antiparallel analogs for this task in the weights of a rich model (Figure A1e). The general benefit of richness together with the presence of conceptual representations continues to align with our theoretical perspective. Across our three visual same-different tasks, we identified generally consistent relationships between learning richness, conceptual solutions, and good performance, supporting our theoretical findings.

## 5 Discussion

We studied equality reasoning using a simple same-different task. We showed that learning richness drives the development of either conceptual or perceptual behavior. Rich MLPs develop conceptual representations, learn from few training examples, and remain largely insensitive to perceptual variation. Meanwhile, lazy MLPs require exhaustive training examples and deteriorate substantially with spurious perceptual changes.

Varying learning richness recapitulates E. Wasserman et al. (2017)'s continuum between perceptual and conceptual behavior on same-different tasks. Perhaps a pigeon's competency at equality reasoning may be broadly comparable to a lazy MLP's, requiring a great deal of training and exhibiting persistent sensitivity to spurious details. Perhaps equality reasoning in human or even language-trained great apes may be comparable to a rich MLP, where learning is faster, less sensitive to spurious details, and presumably involves conceptual abstractions. We suggest that a key parameter underlying these behavioral differences may be learning richness.

Learning richness is a concept imported from machine learning theory, and it is not altogether clear how to measure richness in a living brain. Since richness specifies the degree of change in a neural network's hidden representations, the most direct analogy in the brain is to look for adaptive representations that seem to encode task-specific variables. Such approaches have implicated richness as an essential prop-

erty for context-dependent decision making, multitask cognition, generalizing knowledge, among many other phenomena (Flesch et al., 2022; Ito & Murray, 2023; Johnston & Fusi, 2023; Farrell et al., 2023). Our theory predicts that greater learning richness relates to faster generalization in equality reasoning, and look forward to possible experimental validation of this principle.

Our work also contributes to the longstanding debate on a neural network's facility with abstract reasoning. Rich MLPs demonstrate successful generalization to unseen symbols irrespective of input dimension or even high noise variance. Further, the rich MLP's development of parallel/antiparallel components suggests the formation of abstractions, supporting the account that neural networks may indeed learn to develop and manipulate symbolic representations.

Practically, we demonstrate that learning richness is a vital hyperparameter. Increasing richness generally increases test performance substantially, improves data efficiency, and reduces sensitivity to spurious details. For complex tasks tuned with a large range of $\gamma$, there may be an optimal level of richness. Indeed, for CIFAR-100, we observed that more richness is not always better, and an optimal level exists. We encourage more widespread application of richness parametrizations like $\mu$P, and advocate for adding $\gamma$ to the list of tunable hyperparameters that every practitioner must consider when developing neural networks (Atanasov et al., 2024).

**Acknowledgments.** We thank Hamza Chaudhry, Ben Ruben, Sab Sainathan, Jacob Zavatone-Veth, and members of the Pehlevan Group for many helpful comments and discussions on our manuscript. WLT is supported by a Kempner Graduate Fellowship. CP is supported by NSF grant DMS-2134157, NSF CAREER Award IIS-2239780, DARPA grant DIAL-FP-038, a Sloan Research Fellowship, and The William F. Milton Fund from Harvard University. This work has been made possible in part by a gift from the Chan Zuckerberg Initiative Foundation to establish the Kempner Institute for the Study of Natural and Artificial Intelligence. The computations in this paper were run on the FASRC cluster supported by the FAS Division of Science Research Computing Group at Harvard University.

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

# Appendix

## A Conceptual representations in visual same-different

In Section 4, we experimented with three different visual same-different tasks to validate our theoretical predictions in more complex settings. We found that richer models tend to learn the task with fewer training examples, and display some insensitivity to spurious details. The final signature of a conceptual solution is the presence of conceptual representations. In this appendix, we examine the hidden weights learned by rich and lazy models on these tasks, and present evidence for conceptual representations.

**PSVRT** Recall our MLP given in Eq (1). To interrogate the hidden weights $\mathbf{w}_i$ for conceptual representations, we reshape the weights to match the input shape and visualize them directly. The results are plotted in Figure A1a and b. For ease of visualization, this task consists of images that are two patches wide.

The rich model learns an interesting analog of parallel/antiparallel weights. Recall for our MLPs trained on the simple same-different task, weight vectors associated with negative readouts tend to develop antiparalllel components. Weight vectors associated with positive readouts tend to develop parallel components.

We witness a similar development for PSVRT. For the example weights with a negative readout, adjacent patches are exactly the opposite, learning a negative weight where the neighboring patch has a positive weight. This structure mirrors the antiparallel weights learned in the simple same-different task. One difference is that for PSVRT, while two pairs of regions are precisely the opposite, the other two pairs are the same. While it is impossible to have every region become antiparallel to every other region, it is not obvious why two pairs should become parallel despite the negative readout weight.

Meanwhile, the example weights with a positive readout feature identical patches, matching weights exactly across the four regions of the input. These parallel regions are exactly what we would expect from our consideration of the simple same-different task.

In the lazy regime, the model learns no discernible structure. The magnitudes of both the readout and hidden weights are also significantly smaller. Altogether, the existence of parallel/antiparallel analogs for PSVRT strongly suggests that only the rich model has learned conceptual representations.

**Pentomino** We perform the same analysis of the hidden weights $\mathbf{w}_i$ for the Pentomino task. The results are visualized in Figure A1c and d. For ease of visualization, this task consists of images that are two patches wide.

As we saw for PSVRT, the rich model learns analogs of parallel/antiparallel weights. For the example weights corresponding to a negative readout, the top regions are precisely

the opposite of the bottom regions, suggestive of the antiparallel weight components we characterized in the simple same-different task. For example weights corresponding to a positive readout, all four regions are the same, suggestive of parallel weight components.

These structure emerge only in the rich regime. For the lazy regime model, no discernible structure is learned. The overall magnitudes of the readouts and hidden weights are also much smaller. Altogether, the existence of parallel/antiparallel analogs for Penotmino strongly suggests only the rich model has learned conceptual representations.

### A.1 CIFAR-100

For the CIFAR-100 task, we visualize the hidden weights in the same way as we did for the simple same-different task in Figure 1. We separate the weight vector $\mathbf{w}_i$ into two components, corresponding to the two flattened input images, and measure their alignment. The results are plotted in Figure A1e and f.

For the rich case, alignment associated with negative readouts tends to be negative, and alignment associated with positive readouts tends to be positive, suggestive of the right parallel/antiparallel structure. The alignment is quite similar to what we saw for the rich model on the simple same-different task, though the antiparallel alignment is not as strong. The lazy model shows no apparent correlation at all between readouts and alignment. Altogether, the relationship between readouts and alignment witnessed only in the rich model strongly suggests only the rich model has learned conceptual representations.

## B Hand-crafted solution details

We outline in full detail how our hand-crafted solution solves the same-different task. Recall that the hand-crafted solution is given by the following weight configuration:

$$
\begin{aligned}
\mathbf{w}_1^+ &= (\mathbf{1};\mathbf{1}), \\
\mathbf{w}_2^+ &= (-\mathbf{1};-\mathbf{1}), \\
\mathbf{w}_1^- &= (\mathbf{1};-\mathbf{1}), \\
\mathbf{w}_2^- &= (-\mathbf{1};\mathbf{1}), \\
a^+ &= 1, \\
a^- &= \rho,
\end{aligned}
$$

for $\mathbf{1} = (1,1,\dots,1) \in \mathbb{R}^d$ and some $\rho > 0$. The MLP is given by

$$
\begin{aligned}
f(\mathbf{x}) = a^+ \left( \phi(\mathbf{w}_1^+ \cdot \mathbf{x}) + \phi(\mathbf{w}_2^+ \cdot \mathbf{x}) \right) \\
- a^- \left( \phi(\mathbf{w}_1^- \cdot \mathbf{x}) + \phi(\mathbf{w}_2^- \cdot \mathbf{x}) \right).
\end{aligned}
$$

Upon receiving a *same* example $\mathbf{x}^+ = (\mathbf{z},\mathbf{z})$, our model returns

$$
\begin{aligned}
f(\mathbf{x}^+) &= \phi(\mathbf{1} \cdot \mathbf{z} + \mathbf{1} \cdot \mathbf{z}) + \phi(-\mathbf{1} \cdot \mathbf{z} - \mathbf{1} \cdot \mathbf{z}) \\
&\quad - \rho(\phi(\mathbf{1} \cdot \mathbf{z} - \mathbf{1} \cdot \mathbf{z}) + \phi(-\mathbf{1} \cdot \mathbf{z} + \mathbf{1} \cdot \mathbf{z})) \\
&= |\mathbf{1} \cdot \mathbf{z} + \mathbf{1} \cdot \mathbf{z}| \\
&= 2|\mathbf{1} \cdot \mathbf{z}|,
\end{aligned}
$$

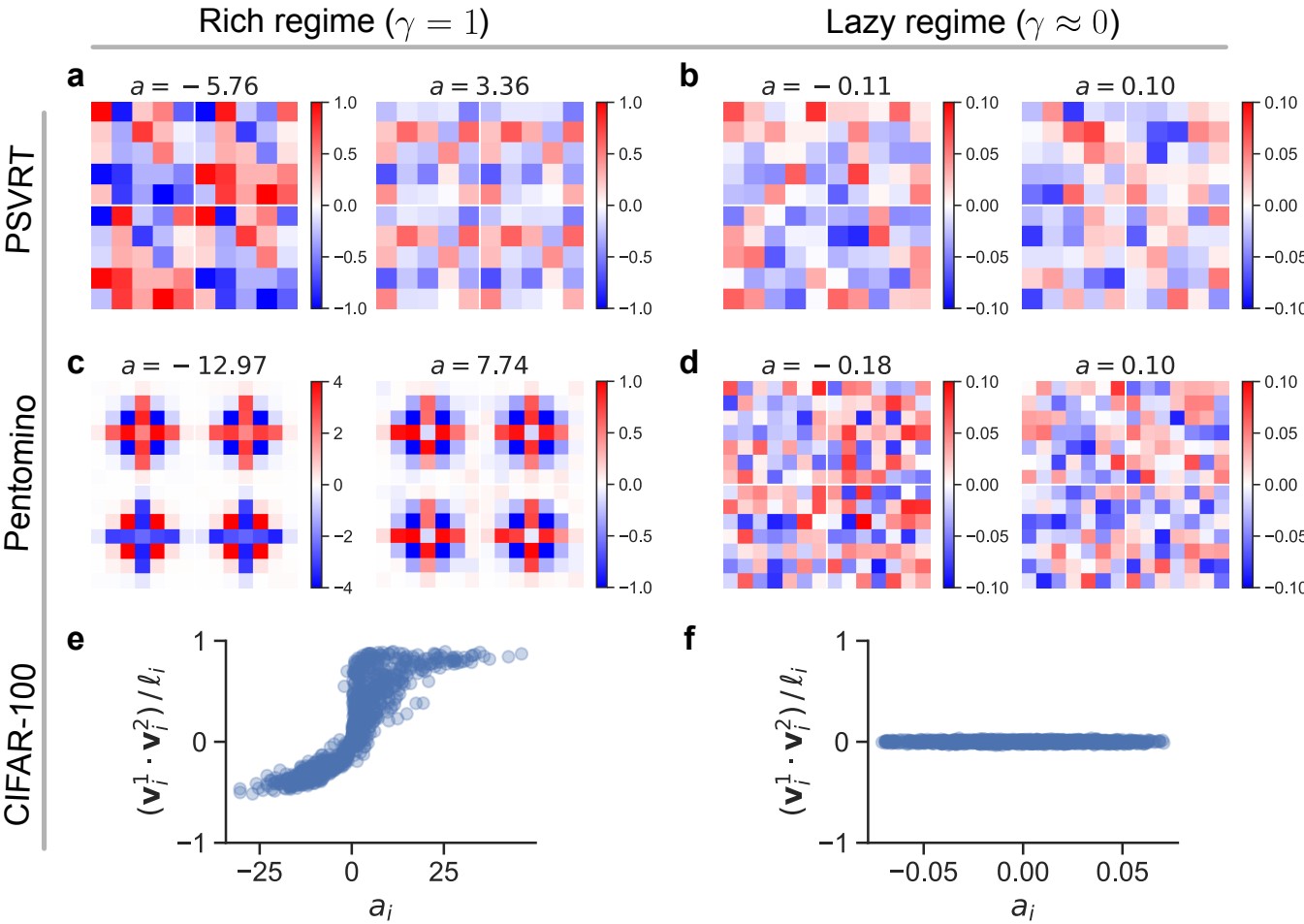

Figure A1: **Conceptual representations in visual same-different tasks. (a)** Visualization of representative hidden weights associated with maximal positive and negative readout weights, for a rich model training on PSVRT. Parallel/antiparallel structures are visible in how regions are either the same or precisely the opposite of their neighbors. **(b)** Visualization of representative hidden weights associated with maximal positive and negative readouts, for a lazy model training on PSVRT. There is no discernible structure, and the magnitudes of both readouts and hidden weights are small. **(c)** The same as (a), but for a rich model on the Pentomino task. Parallel/antiparallel structures are likewise visible. **(d)** The same as (b), but for a lazy model on the Pentomino task. There is no discernible structure, and the magnitudes of both readouts and hidden weights are small. **(e)** For a rich model trained on the CIFAR-100 task, we plot the alignment between weight components corresponding to the two images. There is a distinct parallel/antiparallel-like structure visible in the weight alignment, as we saw for MLPs trained on our simple SD task. **(f)** For a lazy model trained on the CIFAR-100 task, we plot the alignment between weight components corresponding to the two images. There are no discernible parallel/antiparallel structures at all.

which is certainly a positive quantity. Hence, the model classifies all positive examples correctly.

Upon receiving a *different* example $\mathbf{x}^- = (\mathbf{z}, \mathbf{z}')$, our model returns

$$f(\mathbf{x}^-) = \phi(\mathbf{1} \cdot \mathbf{z} + \mathbf{1} \cdot \mathbf{z}') + \phi(-\mathbf{1} \cdot \mathbf{z} - \mathbf{1} \cdot \mathbf{z}')$$
$$- \rho(\phi(\mathbf{1} \cdot \mathbf{z} - \mathbf{1} \cdot \mathbf{z}') + \phi(-\mathbf{1} \cdot \mathbf{z} + \mathbf{1} \cdot \mathbf{z}'))$$

Since training symbols are sampled as $\mathbf{z} \sim \mathcal{N}(0, \mathbf{I}/d)$, we have that $\mathbf{z} \stackrel{d}{=} -\mathbf{z}$. Furthermore, a sum of independent Gaussians remains Gaussian, so $\mathbf{1} \cdot \mathbf{z} \pm \mathbf{1} \cdot \mathbf{z}' \sim \mathcal{N}(0, 2)$. Hence, $f(\mathbf{x}^-) \stackrel{d}{=} u - \rho v$, where $u, v \sim \mathrm{HalfNormal}(0, 2)$. Note, the ReLU nonlinearity ensures these quantities are distributed along a Half-Normal distribution, rather than a Gaussian. Further, $u$ and $v$ are independent since $\mathbf{1} \cdot \mathbf{z} + \mathbf{1} \cdot \mathbf{z}'$ is independent from $\mathbf{1} \cdot \mathbf{z} - \mathbf{1} \cdot \mathbf{z}'$ (the two sums are jointly Gaussian with zero covariance).

The test accuracy of the model on $\mathbf{x}^-$ is given by $p(f(\mathbf{x}^-) < 0)$, which can be expressed as an integral over the joint PDF of $u, v$:

$$p(f(\mathbf{x}^-) < 0) = p(u - \rho v < 0)$$
$$= \frac{1}{2\pi} \int_0^\infty \int_{u/\rho}^\infty \exp\left\{-\frac{u^2 + v^2}{8}\right\} dv\, du.$$

To compute this quantity, we convert to polar coordinates. Let $u = r\cos(\theta)$ and $v = r\sin(\theta)$. Under this change of variables, we have

$$p(f(\mathbf{x}^-) < 0) = \frac{1}{2\pi} \int_{\tan^{-1}(1/\rho)}^{\pi/2} \int_0^\infty r \exp e^{-r^2/8} dr\, d\theta$$
$$= \frac{2}{\pi}\left(\frac{\pi}{2} - \tan^{-1}(1/\rho)\right)$$
$$= \frac{2}{\pi}\tan^{-1}(\rho).$$

For $\rho \to \infty$, this quantity approaches 1. Since $\rho$ cancels in the result for the *same* input, $\rho$ can be arbitrarily large without impacting the classification accuracy on *same* inputs. Hence, the hand-crafted solution overall solves the same-different task provided the relative magnitude of the negative readouts is large.

A technical detail required for both successful positive and negative classifications is that the test example is not precisely orthogonal to the parallel/antiparallel vectors, in which case the relevant dot products would be zero. However, a test example is exactly orthogonal to the weight vectors with probability zero, so this eventuality does not impact the solution's overall test accuracy.

Figure B1 illustrates how parallel/antiparallel weight vectors may correctly classify a *same* or *different* example.

## C  Rich regime details

We conduct our analysis of the rich regime in two parts. We begin with a derivation of the max margin solution to our

Figure B1: **Illustration of hand-crafted solution.** Parallel/antiparallel weight vectors $\mathbf{w}^+, \mathbf{w}^-$ are represented pictorially as sets of two vectors. The dot product operation is represented by conjoining the corresponding vectors: the dot product equals the cosine angle scaled by the magnitudes of the component vectors. For a *same* test example, the $\mathbf{w}^+ \cdot \mathbf{x}$ remains positive while $\mathbf{w}^- \cdot \mathbf{x}$ cancels to zero. For a *different* test example, the relative magnitude $\rho$ enables a successful negative classification.

same-different task in Section C.1. Doing so requires us to replace our model's ReLU activations with quadratic activations. The max margin solution also does not demonstrate the rich model's learning efficiency or insensitivity to perceptual details. To address these shortcomings, we extend our analysis by considering a heuristic construction in which we approximate a rich MLP using an ensemble of independent Markov processes (Section C.2). Using this construction, we derive a finer-grain characterization of the MLP's weight structure, and apply it to estimate the model's test accuracy for varying $L$ and $d$.

### C.1  Max margin solution

An MLP trained on a classification objective often learns a max margin solution over the dataset (Morwani et al., 2023; Chizat & Bach, 2020; Wei et al., 2019). While this outcome is not guaranteed in our setting, studying the structure of the max margin solution nonetheless reveals critical details about how our MLP may be solving the same-different task. Following Morwani et al. (2023), we adopt two conditions to expedite our analysis:

1. We replace a strict max margin objective with a max *average* margin objective over a dataset $\mathcal{D} = \{\mathbf{x}_n, y_n\}_{n=1}^P$

$$\max_\theta \frac{1}{P}\sum_{n=1}^P \left[(2y_n - 1)f(\mathbf{x}_n; \theta)\right],$$

where $\mathbf{x}_n, y_n$ are sampled over a training distribution with $L$ symbols and the objective is subject to some norm constraint on $\theta$. Given the symmetry of the task, a max average margin objective forms a reasonable proxy to the strict max margin.

2. We consider quadratic activations $\phi(\cdot) = (\cdot)^2$ rather than ReLU. Doing so alters our model from Eq (1), but we later use a heuristic construction to argue that the resulting solution is recovered under ReLU activations in a rich learning regime.

We further allow $P, L \to \infty$. Following these simplifications, we derive the max average margin solution.

**Theorem 1.** *Let* $\mathcal{D} = \{\mathbf{x}_n, y_n\}_{n=1}^P$ *be a training set consisting of P points sampled across L training symbols, as specified in Section 2. Let f be the MLP given by Eq 1, with two changes:*

*1. Fix the readouts* $a_i = \pm 1$, *where exactly* $m/2$ *readouts are*

*positive and the remaining are negative.*

*2. Use quadratic activations* $\phi(\cdot) = (\cdot)^2$.

*For weights* $\theta = \{\mathbf{w}_i\}_{i=1}^m$, *define the max margin set* $\Delta(\theta)$ *to be*

$$\Delta(\theta) = \arg\max_\theta \frac{1}{P} \sum_{n=1}^P [(2y_n - 1)f(\mathbf{x}_n; \theta)],$$

*subject to the norm constraints* $||\mathbf{w}_i|| = 1$. *If* $P, L \to \infty$, *then for any* $\mathbf{w}_i = (\mathbf{v}_i^1; \mathbf{v}_i^2) \in \Delta(\theta)$ *and* $\ell_i = ||\mathbf{v}_i^1|| \, ||\mathbf{v}_i^2||$, *we have that* $\mathbf{v}_i^1 \cdot \mathbf{v}_i^2 / \ell_i = 1$ *if* $a_i = 1$ *and* $\mathbf{v}_i^1 \cdot \mathbf{v}_i^2 / \ell_i = -1$ *if* $a_i = -1$. *Further,* $||\mathbf{v}_i^1|| = ||\mathbf{v}_i^2||$.

*Proof.* Let $\mathcal{D}^+$ be the subset of $\mathcal{D}$ consisting of *same* examples and $\mathcal{D}^-$ be the subset of *different* examples. Let $I^+$ be the set of indices $i$ such that the readout weight $a_i > 0$. Let $I^-$ be the set of indices $j$ such that $a_j < 0$. Then our max average margin solution becomes

$$\max_\theta \frac{1}{P} \sum_{n=1}^P [(2y_n - 1)f(\mathbf{x}_n; \theta)] = \max_\theta \frac{1}{P} \left[ \sum_{\mathbf{x}^+ \in \mathcal{D}^+} [f(\mathbf{x}^+; \theta)] - \sum_{\mathbf{x}^- \in \mathcal{D}^-} [f(\mathbf{x}^-; \theta)] \right]$$

$$= \max_\theta \frac{1}{P} \left[ \sum_{\mathcal{D}^+} \left[ \sum_{i \in I^+} |a_i|(\mathbf{w}_i \cdot \mathbf{x}^+)^2 \right] - \sum_{\mathcal{D}^-} \left[ \sum_{i \in I^+} |a_i|(\mathbf{w}_i \cdot \mathbf{x}^-)^2 \right] \right]$$

$$+ \frac{1}{P} \left[ \sum_{\mathcal{D}^-} \left[ \sum_{j \in I^-} |a_j|(\mathbf{w}_j \cdot \mathbf{x}^-)^2 \right] - \sum_{\mathcal{D}^-} \left[ \sum_{j \in I^-} |a_j|(\mathbf{w}_j \cdot \mathbf{x}^+)^2 \right] \right].$$

Suppose we stack all *same* training examples $\mathbf{x}^+$ into a large matrix $\mathbf{X}^+ \in \mathbb{R}^{|\mathcal{D}^+| \times 2d}$ and stack all *different* training examples $\mathbf{x}^-$ into a large matrix $\mathbf{X}^- \in \mathbb{R}^{|\mathcal{D}^-| \times 2d}$. Applying the norm constraints $||\mathbf{w}_i|| = 1$, our max margin solution is resolved by the following objectives

$$\mathbf{w}_*^+ = \arg\max_\mathbf{w} \frac{1}{P} \left[ ||\mathbf{X}^+ \mathbf{w}||^2 - ||\mathbf{X}^- \mathbf{w}||^2 \right] \quad \text{such that } ||\mathbf{w}|| = 1,$$

$$\mathbf{w}_*^- = \arg\min_\mathbf{w} \frac{1}{P} \left[ ||\mathbf{X}^+ \mathbf{w}||^2 - ||\mathbf{X}^- \mathbf{w}||^2 \right] \quad \text{such that } ||\mathbf{w}|| = 1,$$

where $\mathbf{w}_*^+$ and $\mathbf{w}_*^-$ represent the hidden weights of the max average margin solution.

Maximizing (or minimizing) this objective is equivalent to finding the largest (or smallest) eigenvector of the matrix $\mathbf{X} = \frac{1}{P} \left[ (\mathbf{X}^+)^\mathsf{T} \mathbf{X}^+ - (\mathbf{X}^-)^\mathsf{T} \mathbf{X}^- \right]$. In the limit $P, L \to \infty$, this matrix becomes circulant. Let us see how. Note that $\mathbf{X} \in \mathbb{R}^{2d \times 2d}$. Suppose there are exactly $P/2$ *same* examples and $P/2$ *different* examples. Along the diagonal of $\mathbf{X}$ are terms

$$X_{ii} = \frac{1}{P} \sum_{j=1}^{P/2} \left[ (x_{ij}^+)^2 - (x_{ij}^-)^2 \right],$$

where $x_{ij}^+$ corresponds to the $i$th index of the $j$th *same* example, and $x_{ij}^-$ is the same for the $j$th *different* example. Because $L \to \infty$, we have that $x_{i,j}^+ \sim \mathcal{N}(0, 1/d)$ and $x_{i,j}^- \sim \mathcal{N}(0, 1/d)$, where $x_{i,j}^+$ is independent of $x_{i,j}^-$. Hence, $X_{ii} \to 0$ as $P \to \infty$.

Now let us consider the diagonal of the first quadrant

$$X_{i,2i} = \frac{1}{P} \sum_{j=1}^{P/2} \left[ x_{ij}^+ x_{2i,j}^+ - x_{ij}^- x_{2i,j}^- \right].$$

For same examples, $x_{ij}^+ = x_{2i,j}^+$, so

$$\frac{1}{P} \sum_{j=1}^{P/2} x_{ij}^+ x_{2i,j}^+ = \frac{1}{P} \sum_{j=1}^{P/2} \left( x_{ij}^+ \right)^2 \to \frac{1}{2} \mathbb{E} \left[ \left( x_{ij}^+ \right)^2 \right] = \frac{1}{2d}$$

For different examples, $x_{ij}^-$ remains independent of $x_{2i,j}^-$, so

$$\frac{1}{P} \sum_{j=1}^{P/2} x_{ij}^+ x_{2i,j}^+ \to 0.$$

We therefore have overall that $X_{i,2i} \to 1/2d$. The same argument applies for the diagonal of the third quadrant, revealing that $X_{2i,i} \to 1/2d$.

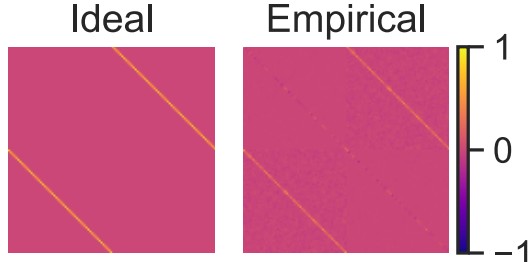

Ideal   Empirical

Figure C1: **Example ideal and empirical X.** The matrix does indeed become circulant with nonzero values on the diagonal of the first and third quadrants. The empirical $\mathbf{X}$ is computed from a batch of 3000 examples sampled from a training set consisting of 64 symbols.

For all other terms $X_{ik}$ where $i \neq k$, $i \neq k/2$, and $i/2 \neq k$, we must have that $x_{ij}$ is independent of $x_{kj}$ for both *same* and *different* examples, so $X_{ik} \to 0$.

Hence, $\mathbf{X}$ is circulant with nonzero values $1/2d$ only on the diagonals of the first and third quadrant. Figure C1 plots an example $\mathbf{X}$. In the remainder of this section, we multiply $\mathbf{X}$ by a normalization factor $2d$. Doing so does not impact the max margin weights, but changes the value along the quadrant diagonals to 1.

The eigendecomposition of a circulant matrix is well studied, and can be given in terms of Fourier modes. In particular, the $\ell$th eigenvector $\mathbf{u}_\ell$ is given by

$$\mathbf{u}_\ell = \frac{1}{\sqrt{2d}}\left(1, r^\ell, r^{2\ell}, \ldots, r^{(2d-1)\ell}\right),$$

where

$$r = e^{\frac{\pi i}{d}}$$

and $\ell$ ranges from 0 to $2d-1$. The corresponding eigenvalues $\lambda_\ell$ are

$$\lambda_\ell = r^{d\ell} = e^{\ell\pi i}.$$

This expression implies that $\lambda_\ell = 1$ for even $\ell$ and $\lambda_\ell = -1$ for odd $\ell$. Hence, $\mathbf{w}_*^+$ lies in the subspace spanned by $\mathbf{u}_\ell$ for even $\ell$, and $\mathbf{w}_*^-$ lies in the subspace spanned by eigenvectors with odd $\ell$.

To characterize this solution further, suppose we partition a weight vector $\mathbf{w} \in \mathbb{R}^{2d}$ into equal halves $\mathbf{w} = (\mathbf{v}^1; \mathbf{v}^2)$, where $\mathbf{v}^1 \in \mathbb{R}^d$. Considering the even case first, suppose $\mathbf{w} \in \mathbf{U}_2$ where $\mathbf{U}_2 = \mathrm{span}\left\{\mathbf{u}_0, \mathbf{u}_2, \ldots, \mathbf{u}_{2(d-1)}\right\}$. Then there exist coefficients $c_0, c_2, \ldots c_{2(d-1)}$ such that

$$\mathbf{w} = \sum_{n=0}^{d-1} c_{2n}\mathbf{u}_{2n}.$$

Note that

$$r^{k\ell_1} \cdot \overline{r}^{k\ell_2+d} = e^{\frac{k\pi i}{d}(\ell_1-\ell_2)} \cdot e^{-\pi i \ell_2}.$$

If we partition our set of eigenvectors as $\mathbf{u}_\ell = (\mathbf{s}_\ell^1, \mathbf{s}_\ell^2)$, then

$$\mathbf{s}_{\ell_1}^1 \cdot \mathbf{s}_{\ell_2}^2 = e^{-\pi i \ell_2} \sum_{k=0}^{d-1} e^{\frac{k\pi i}{d}(\ell_1-\ell_2)}.$$

This quantity is 0 when $\ell_1 \neq \ell_2$. Otherwise, it is 1 if $\ell_1 = \ell_2$ are even and $-1$ if they are odd. Hence, for $(\mathbf{v}^1; \mathbf{v}^2) \in \mathbf{U}_2$, we have that

$$\mathbf{v}^1 \cdot \mathbf{v}^2 = \frac{1}{2d}\left(c_0^2 + c_2^2 + \ldots + c_{2(d-1)}^2\right).$$

Observe also that

$$||\mathbf{v}^1|| = ||\mathbf{v}^2|| = \frac{1}{\sqrt{2d}}\sqrt{c_0^2 + c_2^2 + \ldots + c_{2(d-1)}^2},$$

so we must have

$$\frac{\mathbf{v}^1 \cdot \mathbf{v}^2}{||\mathbf{v}^1||\,||\mathbf{v}^2||} = 1.$$

In this way, we see that the components of $\mathbf{w}_*^+$ must be *parallel* and share the same mangitude. We may repeat the same calculation for $\mathbf{w} \in \mathbf{U}_1$, where $\mathbf{U}_1 = \mathrm{span}\{\mathbf{u}_1, \mathbf{u}_3, \ldots, \mathbf{u}_{2d-1}\}$. Doing so reveals that

$$\frac{\mathbf{v}^1 \cdot \mathbf{v}^2}{||\mathbf{v}^1||\,||\mathbf{v}^2||} = -1,$$

so the components of $\mathbf{w}_*^-$ must be *antiparallel*. $\square$

### C.2 Heuristic construction

By examining the max average margin solution, we witness the emergence of parallel/antiparallel weight vectors. In Section 3.1, we discussed how parallel/antiparallel weights allow an MLP to solve the SD task. However, it remains unclear how to characterize the learning efficiency and insensitivity to spurious perceptual details in the resulting model, and whether these results apply at all to a ReLU MLP trained on a finite dataset. To begin answering these questions, we develop a heuristic construction that summarizes the learning dynamics of a ReLU MLP over the subsequent sections. We will demonstrate that

1. The hidden weights $\mathbf{w}$ become parallel (or antiparallel) for correspondingly positive (or negative) readout weights $a$

2. The magnitude of the readout weights are such that $|\overline{a^-}| > |\overline{a^+}|$, where $\overline{a^-}$ denotes the average across negative readout weights and $\overline{a^+}$ denotes the average across positive readout weights

We then leverage our understanding of the weight structure to estimate a rich model's test accuracy on our SD task. The remainder of this appendix is dedicated to developing this heuristic approach.

We proceed using a Markov process approximation to the full learning dynamics in the noiseless setting ($\sigma^2 = 0$). Observe that the gradient updates to the readout and hidden weights take the following form. For a batch containing $N$ training examples,

$$\Delta a_i = -\frac{c}{N}\sum_{j=1}^{N}\frac{\partial \mathcal{L}_j}{\partial f}\phi(\mathbf{w}_i \cdot \mathbf{x}_j),$$

$$\Delta \mathbf{w}_i = -\frac{c}{N}\sum_{j=1}^{N}\frac{\partial \mathcal{L}_j}{\partial f}a_i\phi'(\mathbf{w}_i \cdot \mathbf{x}_j)\mathbf{x}_j,$$

where $c = \frac{\alpha}{\gamma\sqrt{d}}$, $\alpha$ is the learning rate, and

$$-\frac{\partial L_j}{\partial f} = -\frac{\partial L(y,f(\mathbf{x}))}{\partial f}\bigg|_{y_j,f(\mathbf{x}_j)}$$
$$= \frac{y_j}{1+e^{f(\mathbf{x}_j)}} - \frac{1-y_j}{1+e^{-f(\mathbf{x}_j)}}. \quad \text{(C.1)}$$

Focusing on $\Delta\mathbf{w}_i$, we rewrite its gradient update as

$$\Delta\mathbf{w}_i = \sum_{j=1}^N \xi_{ij}\mathbf{x}_j\,,$$

where

$$\xi_{ij} = -\frac{c}{N}\frac{\partial L_j}{\partial f} a_i \phi'(\mathbf{w}_i\cdot\mathbf{x}_i)\,.$$

When written in this form, it becomes clear that the hidden weight gradient updates lie in the basis of the training examples $\mathbf{x}_j$. If the initialization of the hidden weights $\mathbf{w}_i$ is small, then $\mathbf{w}_i$ lies approximately in the basis of training examples also. Specifically, we require that

$$\mathbf{w}_i(0) \neq \mathbf{0} \text{ and } \frac{1}{\xi_{ij}}\mathbf{w}_i(0)\cdot\mathbf{x}_j \to 0 \quad \text{as } d\to\infty, \quad \text{(C.2)}$$

where $\mathbf{w}_i(0)$ refers to $\mathbf{w}_i$ at initialization (generally, $\mathbf{w}_i(t)$ is the value of $\mathbf{w}_i$ after $t$ gradient steps). The requirement that $\mathbf{w}_i(0) \neq \mathbf{0}$ ensures that the initial gradient update is nonzero.

Suppose our training set consists of $L$ symbols $\mathbf{z}_1; \mathbf{z}_2,\ldots,\mathbf{z}_L$. If we partition $\mathbf{w}_i = (\mathbf{v}_i^1; \mathbf{v}_i^2)$, then after $t$ gradient steps and in the infinite limit $d\to\infty$

$$\mathbf{v}_i^1(t) = \omega_{i,1}^1(t)\mathbf{z}_1 + \omega_{i,2}^1(t)\mathbf{z}_2 + \ldots + \omega_{i,L}^1(t)\mathbf{z}_L\,,$$
$$\mathbf{v}_i^2(t) = \omega_{i,1}^2(t)\mathbf{z}_1 + \omega_{i,2}^2(t)\mathbf{z}_2 + \ldots + \omega_{i,L}^2(t)\mathbf{z}_L\,, \quad \text{(C.3)}$$

where $\omega_{i,k}^p(t)$ corresponds to the overlap $\mathbf{w}_i^p(t)\cdot\mathbf{z}_k$.[6] Note, for these relations to hold, we require that $\mathbf{z}_i\cdot\mathbf{z}_j = \delta_{ij}$ as $d\to\infty$. In this way, we may consider the $\omega$'s to be the coordinates of $\mathbf{w}$ in the basis of the training symbols $\mathbf{z}$.

Note that $\omega$ is a function of the update coefficients $\xi_{ij}$. If $\mathbf{w}_i\cdot\mathbf{x}_j < 0$, then $\xi_{ij} = 0$. Otherwise, $\xi_{ij}$ depends on $\frac{\partial L_j}{\partial f}$ and $a_i$, introducing many additional and complex couplings to other parameters in the model. Our ultimate goal is to understand the general structure of the hidden weights $\mathbf{w}_i$, rather than to obtain exact formulas, so we apply the following coarse approximation

$$\xi_{ij} = \begin{cases} \text{sign}\left(-\frac{\partial L_j}{\partial f} a_i\right) & \mathbf{w}_i\cdot\mathbf{x}_j > 0\,, \\ 0 & \mathbf{w}_i\cdot\mathbf{x}_j \leq 0\,. \end{cases} \quad \text{(C.4)}$$

Such an approximation resembles sign-based gradient methods like signSGD (Bernstein et al., 2018) and Adam (Kingma, 2014). We also verify empirically that this approximation describes rich regime learning dynamics well.

---

[6]The large number of indices on $\omega$ is unwieldy, so we will omit some or all of them where context allows.

From Eq (C.1), observe that $\text{sign}\left(-\frac{\partial L_j}{\partial f}\right) = 1$ given a label $y_j = 1$, and $\text{sign}\left(-\frac{\partial L_j}{\partial f}\right) = -1$ for the label $y_j = 0$. Recall from our hand-crafted solution that $a_i > 0$ implies that $\mathbf{w}_i$ should align with *same* examples, and $a_i < 0$ implies that $\mathbf{w}_i$ should align with *different* examples. Hence, if $\mathbf{w}_i\cdot\mathbf{x}_j > 0$, we conclude that $\xi_{ij} = 1$ if the example $\mathbf{x}_j$ matches the corresponding readout weight $a_i$ — that is, $\mathbf{x}_j$ is *same* and $a_i > 0$, or $\mathbf{x}_j$ is *different* and $a_i < 0$. Otherwise, if there is a mismatch, $\xi_{ij} = -1$. In this way, we may interpret $\mathbf{w}_i$ as a "state vector" to which we add or subtract examples $\mathbf{x}_j$ based on a simple set of update rules. We proceed to study the limiting form of $\mathbf{w}_i$ by treating it as a Markov process. Our approximation in Eq (C.4) decouples the dependency between hidden weight vectors. The set of hidden weights can be treated as an ensemble of independent Markov processes evolving in parallel, allowing us to understand the overall structure of the hidden weights.

## C.3 Markov process approximation

Altogether, we summarize the learning dynamics on $\mathbf{w}$ through the following Markov process. In the remainder of this section, we drop the index $i$ from $\mathbf{w}_i$ and $a_i$, and $\mathbf{w}$ and $a$ should be understood as a representative sets of weights. Similarly, we write $\omega_k^p(t)$ to represent the coefficient of $\mathbf{v}^p$ (the $p$th partition of $\mathbf{w}$, $p \in \{1,2\}$) for the $k$th training symbol after $t$ steps. The set of coefficients $\omega$ represent the state of the Markov process, which proceeds as follows.

**Step 1.** Initialize $\omega_k^p(0) = 0$ for all $k$, $p$. Initialize the time step $t = 0$. Initialize batch updates $b_k^p = 0$ and batch counter $n = 0$. Set the batch size $N$.

**Step 2.** Sample an integer $u$ uniformly at random from the set $[L] = \{1,2,\ldots,L\}$.
   With probability $1/2$, set $v = u$.
   Otherwise, sample $v$ uniformly from $[L]\setminus\{u\}$.

**Step 3.** Compute $\rho = \omega_u^1(t) + \omega_v^2(t)$.
   If $\rho > 0$, proceed to step 4.
   If $\rho = 0$, with probability $1/2$ proceed to step 4. Otherwise, proceed to step 5.
   If $\rho < 0$, proceed to step 5.

**Step 4.** If $a > 0$, $u = v$, or $a < 0$, $u \neq v$, update

$$b_u^1 \leftarrow b_u^1 + 1\,,$$
$$b_v^2 \leftarrow b_v^2 + 1\,.$$

Otherwise, update

$$b_u^1 \leftarrow b_u^1 - 1\,,$$
$$b_v^2 \leftarrow b_v^2 - 1\,.$$

**Step 5.** Increment the batch counter $n \leftarrow n+1$.

If $n = N$, Set $\omega_k^p(t+1) \leftarrow \omega_k^p(t) + b_k^p$ and increment the time step $t \leftarrow t+1$. Reset $n \leftarrow 0, b_k^p \leftarrow 0$.

Proceed to step 2.

**Remarks.** This Markov process approximates the learning dynamics of an MLP given the simplification in Eq (C.4). We elaborate on the link below.

In step 1, we initialize the weight vector to zero. In practice, weight vectors are initialized as $\mathbf{w} \sim \mathcal{N}(\mathbf{0}, \mathbf{I}/d)$. For large $d$, the condition required in Eq (C.2) allows us to approximate this as zero, with some caveats described below.

In step 2, we sample a training example. Because we operate in the basis spanned by training examples, we discard the vector content of a training symbol and consider only its index. With half the training examples being *same* and half being *different*, we sample indices accordingly.

In step 3, we consider the overlap $\mathbf{w} \cdot \mathbf{x}$, where $\mathbf{x} = (\mathbf{z}_u, \mathbf{z}_v)$. Given the assumptions in Eq (C.2), we have that

$$\begin{aligned}
\rho &= \mathbf{w} \cdot \mathbf{x} \\
&= \omega_u^1 ||\mathbf{z}_u||^2 + \omega_v^2 ||\mathbf{z}_v||^2 \\
&= \omega_u^1 + \omega_v^2.
\end{aligned}$$

If the overlap $\rho$ is positive, then the update coefficient $\xi \neq 0$, so we branch to the step where the state is updated. If $\rho$ is negative, we must have that $\xi = 0$, so we skip the update. If $\omega_u^1 + \omega_v^2 = 0$, the overlap still picks up the initialization, $\mathbf{w} \cdot \mathbf{x} = \mathbf{w}(0) \cdot \mathbf{x}$. In the limit $d \to \infty$, we have that $\mathbf{w}(0) \cdot \mathbf{x} \to 0$. However, $\mathbf{w}(0) \cdot \mathbf{x} \neq 0$ almost surely for any finite $d$. Because $\mathbf{w}(0)$ and $\mathbf{x}$ are radially symmetric about the origin, their overlap is positive with probability 1/2. Thus, when $\rho = 0$, we branch to the corresponding positive or negative overlap steps each with probability 1/2.

In step 4, we apply the updates from $\xi = \pm 1$, based on whether the training example $\mathbf{z}_u, \mathbf{z}_v$ matches the readout weight $a$. We conclude the training loop in step 5, and restart with a fresh example.

## C.4 Limiting weight structure

With the setup now complete, we analyze the Markov process proposed above to understand the limiting structure of $\mathbf{w}$ and $a$. Specifically, we will show that for large $L$ and as $t \to \infty$,

1. If $a > 0$, then $||\mathbf{v}_1|| = ||\mathbf{v}_2||$ and $\mathbf{v}^1 \cdot \mathbf{v}^2/(||\mathbf{v}^1|| \, ||\mathbf{v}^2||) = 1$

2. If $a < 0$, then $||\mathbf{v}_1|| = ||\mathbf{v}_2||$ and $\mathbf{v}^1 \cdot \mathbf{v}^2/(||\mathbf{v}^1|| \, ||\mathbf{v}^2||) = -1$

3. Suppose $\overline{a^+}$ is the average over all weights $a > 0$, and $\overline{a^-}$ is the average over all weights $a < 0$. Then $|\overline{a^-}| > |\overline{a^+}|$.

Our general approach will involve factorizing the Markov process into an ensemble of random walkers with simple dependencies, then reason about the long time-scale behavior of these walkers. For simplicity, we will focus on the single batch case $N = 1$. Generalizing to $N > 1$ is straightforward but notationally cluttered, and does not change the final result.

**C.4.1 *Same* case** We begin by examining weights $\mathbf{w}$ such that the corresponding readout $a > 0$. Recall that these weights favor *same* examples. Consider a random walker on $\mathbb{R}$ whose position at time $t$ is given by $s_u(t) = \omega_u^1(t) + \omega_u^2(t)$. Then the following rules govern the walker's dynamics:

1. If $s_u(t) > 0$ and the model receives a *same* training example $(\mathbf{z}_u, \mathbf{z}_u)$, then $s_u(t+1) = s_u(t) + 2$.

2. If the model receives a *different* training example $(\mathbf{z}_u, \mathbf{z}_v)$, where $u \neq v$ then $s_u(t+1) \geq s(t) - 1$.

3. If $s_u(T) < 0$, then $s_u(t) < 0$ for all $t > T$.

Rules 1 and 2 reflect the update dynamics of the Markov process. Since $s_u(t) > 0$, upon receiving a *same* example $(\mathbf{z}_u, \mathbf{z}_u)$, we witness updates $\omega_u^1(t+1) = \omega_u^1(t) + 1$ and $\omega_u^2(t+1) = \omega_u^2(t) + 1$, so $s_u(t+1) = s_u(t) + 2$. Similarly, upon receiving a *different* example $(\mathbf{z}_u, \mathbf{z}_v)$, we have that $\omega_u^1(t+1) = \omega_u^1(t) - 1$ if $\omega_u(t) + \omega_v(t) > 0$, so $s_u$ decreases at most by 1. Finally, for rule 3, if $s_u$ ever falls below 0, then it will never increment. Hence, $s_u$ will remain negative for all subsequent steps.

Together, these rules partition our ensemble of walkers $s_u$ into two sets: walkers with positive position $\mathcal{S}^+(t) = \{s_u : s_u(t) > 0\}$ and walkers with negative position $\mathcal{S}^-(t) = \{s_v : s_v(t) < 0\}$. We will show that under typical conditions, members of $\mathcal{S}^+$ grow continually more positive, while members of $\mathcal{S}^-$ grow continually more negative. We denote $n^+(t) = |\mathcal{S}^+(t)|$ and $n^-(t) = |\mathcal{S}^-(t)|$. Where the meaning is unambiguous, we drop the indices $t$.

We first make the following counterintuitive observation about the relative occurrence of *same* and *different* examples. Although training examples are sampled from each class with equal probability, the probabilities of observing a *same* or *different* pair when conditioned on observing a particular training symbol are *not* equal. Suppose we would like to count all training pairs that contain at least one occurrence of $\mathbf{z}_u$. Out of all *same* examples, we would expect roughly $\frac{1}{L}$ of such examples to contain $\mathbf{z}_u$. Out of all *different* examples, we would expect roughly $\frac{2}{L(L-1)}$ occurrences of the pair $(\mathbf{z}_u, \mathbf{z}_v)$, for a specific $v \neq u$. Across all $L - 1$ possible $v$, this proportion rises to $\frac{2}{L(L-1)} \cdot (L-1) = \frac{2}{L}$. Hence, the probability of observing a *same* example conditioned on containing $\mathbf{z}_u$ is actually $\frac{1/L}{1/L + 2/L} = \frac{1}{3}$, while the probability of observing *different* is $\frac{2}{3}$.

Suppose we allow our Markov process to run for $t$ time steps, after which there are $n^+$ walkers in a positive position and $n^-$ walkers in a negative position, among $L = n^+ + n^- \equiv n$ total walkers. Upon receiving the next training example $(\mathbf{z}_u, \mathbf{z}_v)$, there are four possible outcomes.

**Case 1.** The walker $s_u(t) > 0$ and we receive a *same* example (so $u = v$). In this case, $s_u(t+1) = s_u(t) + 2$. The probability of observing a *same* pair containing $\mathbf{z}_u$ is $\frac{1}{3}$, so we summarize this case as

$$p(s_u \leftarrow s_u + 2 \mid s_u > 0) = \frac{1}{3}. \tag{C.5}$$

**Case 2.** The walker $s_u(t) > 0$ and we receive a *different* example (so $u \neq v$). Whether $s_u$ decrements in this case is complex to determine, and depends on the precise coordinates $\omega_u$ and $\omega_v$. We treat this issue coarsely by modeling the probability of decrement through an average case approximation: if $s_v > 0$, we assume that $s_u$ will always decrement; if $s_v < 0$, we assume that $s_u$ will decrement with some mean probability $\mu$. Since $p(s_v > 0) = \frac{n^+ - 1}{n-1}$ and $p(s_v < 0) = \frac{n^-}{n-1}$, and the probability of selecting a *different* example overall remains $2/3$, we summarize this case as

$$p(s_u \leftarrow s_u - 1 \,|\, s_u > 0) = \frac{2}{3} \left( \frac{n^+ - 1}{n-1} + \frac{n^-}{n-1}\mu \right). \quad \text{(C.6)}$$

This average case approximation is similar in flavor to the mean field *ansatz* common in physics, and we employ it for similar reasons: it simplifies a complex many-bodied interaction into a simple interaction between a single body and an average field. We validate the accuracy of this approximation later below.

**Case 3.** The walker $s_u(t) < 0$ and we receive a *same* example. No updates occur in this case. For completeness, we summarize it as

$$p(s_u \leftarrow s_u + 2 \,|\, s_u < 0) = 0. \quad \text{(C.7)}$$

**Case 4.** The walker $s_u(t) < 0$ and we receive a *different* example. We again apply a coarse, average case approximation to model the probability of decrementing. If $s_v < 0$, we assume that $s_u$ will never decrement. If $s_v > 0$, the probability of decrementing again depends on our mean quantity $\mu$. The probability of selecting a *different* example overall remain $2/3$, so we summarize this case as

$$p(s_u \leftarrow s_u - 1 \,|\, s_u < 0) = \frac{2}{3} \left( \frac{n^+}{n-1}\mu \right). \quad \text{(C.8)}$$

To gain greater insight into $\mu$, we consider how much a walker's position may drift as it encounters different training examples. Define a walker's *expected drift* to be the quantity $\Delta s(t) = \mathbb{E}[s(t+1) - s(t)]$, averaged over possible walker states $s$. Then under Eq (C.5) and Eq (C.6), considering a positive walker $s^+ > 0$

$$\Delta s^+(t) = 2\, p(s^+ > 0)\, p(s^+ \leftarrow s^+ + 2 \,|\, s^+ > 0)$$
$$- p(s^+ > 0)p(s^+ \leftarrow s^+ - 1 \,|\, s^+ > 0)$$
$$= \frac{2n^+}{3n} \left( 1 - \frac{n^+ + n^-\mu - 1}{n-1} \right)$$
$$= \frac{2n^+ n^- (1-\mu)}{3n(n-1)}. \quad \text{(C.9)}$$

Similarly, under Eq (C.7) and Eq (C.8), a negative walker $s^-$ has expected drift

$$\Delta s^-(t) = -\, p(s^- < 0)\, p(s^- \leftarrow s^- - 1 \,|\, s^- < 0)$$
$$= -\frac{2n^- n^+ \mu}{3n(n-1)}. \quad \text{(C.10)}$$

Suppose $\mu = 1$. In this case, if we encounter a *different* example $(\mathbf{z}_u, \mathbf{z}_v)$ such that $s_u > 0$ and $s_v < 0$, then $s_u$ will always decrement. On average, $\Delta s_u = 0$ while $\Delta s_v$ is a negative quantity, indicating that $s_u$ will on average remain around the same position while $s_v$ decreases. However, if $s_v$ decreases without bound, there comes a point where $\omega_u + \omega_v < 0$, preventing further decrements. This situation implies that $\mu = 0$, resulting in $\Delta s_u > 0$ and $\Delta s_v = 0$. However, if $s_u$ now increases without bound, there comes a point where $\omega_u + \omega_v > 0$, allowing again further decrements, raising our mean update probability back to $\mu = 1$.

In general, if $|\Delta s_u| < |\Delta s_v|$, we experience further increments until $|\Delta s_u| > |\Delta s_v|$, at which point we experience further decrements, returning us back to $|\Delta s_u| < |\Delta s_v|$. Over a long time period, we might therefore expect our dynamics to settle around an average point $|\Delta s_u| = |\Delta s_v|$. If this is true, then we employ the relation $|\Delta s^+| = |\Delta s^-|$ as a self-consistency condition to solve for $\mu$. Equating (C.9) and (C.10) reveals that $\mu = \frac{1}{2}$.

Altogether, we arrive at the following picture of the walkers' dynamics. Walkers $s_u$ at a positive position drift with an average rate

$$\Delta s_u = \frac{n^+ n^-}{3n(n-1)}. \quad \text{(C.11)}$$

Meanwhile, walkers $s_v$ at a negative position drift with an average rate $\Delta s_v = -\Delta s_u$. Over long time periods, $s_u \to \infty$ while $s_v \to -\infty$. Because positive updates increment both coordinates $\omega_u^1$ and $\omega_u^2$ equally, we have that $\omega_u^1 \approx \omega_u^2 > 0$. Meanwhile, because negative updates have a higher chance of decrementing a more positive coordinate, we also have that $\omega_v^1 \approx \omega_v^2 < 0$. In this way, we must have overall that $||\mathbf{v}_1|| = ||\mathbf{v}_2||$ and $\mathbf{v}^1 \cdot \mathbf{v}^2/(||\mathbf{v}^1||\, ||\mathbf{v}^2||) = 1$ at long time scales, confirming that a weight vector aligned with *same* examples adopts parallel components.

To validate our key assumption on $\mu$, we simulate the Markov process $100$ times with $L = 16$ and a batch size of $512$. We empirically find that $\mu = 0.508 \pm 0.056$ (given by two standard deviations), matching closely our conjecture that $\mu = \frac{1}{2}$.

One caveat we have not addressed is the case where $n^- = n$. In this case, no further updates occur and the weights are frozen in their current position. However, as we suggest later in Section C.4.3, the corresponding readout of these weights will be relatively small, reducing its impact. In practice, "dead" weights like these that are negatively aligned with all training symbols appear to be rare in trained models.

**C.4.2 *Different* case** For weights $\mathbf{w}$ corresponding to a readout $a < 0$, similar rules hold but now with flipped signs. Considering again a random walker with position $s_u(t) = \omega_u^1(t) + \omega_u^2(t)$, the following rules govern the walker's dynamics:

1. If $s_u(t) > 0$ and the model receives a training example $(\mathbf{z}_u, \mathbf{z}_u)$, then $s_u(t+1) = s_u(t) - 2$.

2. If the model receives a training example $(\mathbf{z}_u, \mathbf{z}_v)$ or its reverse $(\mathbf{z}_v, \mathbf{z}_u)$, where $u \neq v$ then $s_u(t+1) \leq s(t) + 1$.

The rules follow from the update dynamics of the Markov process precisely as before, now for weights sensitive to *different* examples. Note, there is no equivalence to Rule 3 in this case, since a walker may (in most cases) continue to receive either positive or negative updates regardless of the sign of its position. Indeed, this added symmetry to the *different* case simplifies the analysis somewhat compared to the *same* case, where it was necessary to study the interactions between two ensembles of random walkers that evolve in different ways. Here, we may treat all walkers uniformly.

Our general approach for analyzing this case is the same. Conditioned on training examples containing the $u$th training symbol, recall that observing a *same* pair $(\mathbf{z}_u, \mathbf{z}_u)$ has probability 1/3, and observing a *different* pair $(\mathbf{z}_u, \mathbf{z}_v)$ has probability 2/3. Then we have the following cases:

**Case 1.** The walker $s_u(t) > 0$ and receives a *same* example. In this case, $s_u(t+1) = s_u(t) - 2$. The probability of observing a *same* pair containing $\mathbf{z}_u$ is 1/3, so we summarize this case as

$$p(s_u \leftarrow s_u - 2 \,|\, s_u > 0) = \frac{1}{3}. \tag{C.12}$$

**Case 2.** The walker $s_u(t) > 0$ and we receive a *different* example. Whether $s_u$ increments is complex to determine, and depends on the precise coordinates $\omega_u$ and $\omega_v$. As before, we treat this issue coarsely by approximating the probability of incrementing through an average case parameter $\mu$. The probability of selecting a *different* example overall remains $2/3$, so we summarize this case as

$$p(s_u \leftarrow s_u + 1 \,|\, s_u > 0) = \frac{2}{3}\mu. \tag{C.13}$$

**Case 3.** The walker $s_u(t) < 0$ and we receive a *same* example. No updates occur in this case. For completeness, we summarize it as

$$p(s_u \leftarrow s_u - 2 \,|\, s_u < 0) = 0. \tag{C.14}$$

**Case 4.** The walker $s_u(t) < 0$ and we receive a *different* example. We again use our average case parameter to describe the probability of incrementing. The probability of selecting a *different* example overall remains $2/3$, so we summarize this case as

$$p(s_u \leftarrow s_u + 1 \,|\, s_u < 0) = \frac{2}{3}\mu. \tag{C.15}$$

To obtain a self-consistent condition for $\mu$, we consider again the expected drift of walkers at positive or negative positions. After $t$ timesteps have elapsed, suppose the number of walkers with position $s^+ > 0$ is $n^+$, and the number of walkers with position $s^- < 0$ is $n^-$, where $L = n^+ + n^- \equiv n$. Then

combining Eq (C.12) and (C.13), the expected drift for positive walkers is

$$\Delta s^+ = \frac{2n^+}{3n}(\mu - 1). \tag{C.16}$$

Combining Eq (C.14) and (C.15), the expected drift of the negative walkers is

$$\Delta s^- = \frac{2n^-}{3n}\mu \tag{C.17}$$

Note, unlike the *same* case, the expected drift for positive walkers is *negative*, and the expected drift for negative walkers is *positive*. Hence, for $\mu \in (0, 1)$, if $|\Delta s^+| > |\Delta s^-|$, the number of positive walkers decreases faster than it increases, so we eventually reach a point where $|\Delta s^+| \leq |\Delta s^+|$. However, when $|\Delta s^+| < |\Delta s^-|$, the number of negative walkers decreases faster than it increases, so we oscillate back to $|\Delta s^+| \geq |\Delta s^-|$. Over a long time period, we assume our walkers settle around an average point $|\Delta s^+| = |\Delta s^-|$. If this is true, then as before, we employ the relation $|\Delta s^+| = |\Delta s^-|$ as a self-consistency condition to solve for $\mu$. Equating (C.16) and (C.17) indicates that $\mu = \frac{n^+}{n}$.

There are three potential settings for $n^+$ to consider: $0 < n^+ < n$, $n^+ = n$, or $n^+ = 0$. Let us begin with $0 < n^+ < n$. Because $n^+ < n$, the average position of positive walkers experiences a net negative drift, gradually bringing them closer to zero. Because $n^+ > 0$, the average position of negative walkers experiences a net positive drift, gradually bringing them closer to zero also. Over a long period of time, we would therefore expect $n^+ \approx n^-$ (and $\mu \approx \frac{1}{2}$).

If $n^+ = n$, then all walkers are in a positive position and $\Delta s^+ = 0$. Over a long period of time, as the variance in walker position grows, through random chance at least one walker will eventually drift to a negative position, returning us to the case where $0 < n^+ < n$. If $n^+ = 0$, then all walkers are in a negative position, and $\Delta s^- > 0$. Over time, the walkers' average position grows more positive, until at least one becomes positive and we again re-enter the case $0 < n^+ < n$.

Altogether, we arrive at the following picture of the walker's dynamics. Walkers at a positive position drift with a negative rate down to zero, and walkers at a negative position drift with a positive rate up to zero. After sufficient time has elapsed, we would therefore expect the position of all walkers to be close to zero. However, for a walker $s_u$, positive updates increment the underlying coordinates $\omega_u^1$ and $\omega_u^2$ asymmetrically. Furthermore, a more positive coordinate receives a positive update with greater probability. Hence, we must have that $\omega_u^1 \gg \omega_u^2$ or $\omega_u^1 \ll \omega_u^2$. Because their sum must remain close to zero, it must be true that $\omega_u^1 \approx -\omega_u^2$. Thus, we have overall that $||\mathbf{v}_1|| = ||\mathbf{v}_2||$ and $\mathbf{v}^1 \cdot \mathbf{v}^2 / (||\mathbf{v}^1|| \, ||\mathbf{v}^2||) = -1$, confirming that a weight vector aligned with *different* examples adopts antiparallel components.

To validate our key assumption on $\mu$, we simulate the Markov process 100 times with $L = 16$ and a batch size of 512. We empirically find that $\mu = 0.499 \pm 0.009$ (given by two standard deviations), matching closely our conjecture that $\mu = \frac{1}{2}$.

**C.4.3 Magnitude of readouts** The final piece to demonstrate in our study of the rich regime is that $|\overline{a^+}| < |\overline{a^-}|$, where $\overline{a^+}$ corresponds to the average across all positive readout weights and $\overline{a^-}$ corresponds to the average across all negative readout weights. Exactly characterizing these magnitudes is difficult, so we apply a heuristic argument based what we learned about the structure of parallel and antiparallel weights above and support it with numeric evidence.

Recall that the update rule for a readout weight $a_i$ is given by

$$\Delta a = -\frac{c}{N} \sum_{j=1}^{N} \frac{\partial \mathcal{L}_j}{\partial f} \phi(\mathbf{w} \cdot \mathbf{x}_j),$$

where $c = \frac{\alpha}{\gamma \sqrt{d}}$ and $\alpha$ is the learning rate. Suppose $\mathbf{x}_j = (\mathbf{z}_u, \mathbf{z}_v)$. Then the update rule becomes

$$\Delta a = -\frac{c}{N} \sum_{u,v} \frac{\partial \mathcal{L}_{u,v}}{\partial f} \phi(\omega_u^1 + \omega_v^2).$$

If $\frac{\partial \mathcal{L}}{\partial f}$ is about the same in magnitude across all training examples, then our readout updates are proportional to

$$\Delta a \propto \sum_{u,v} S(u,v) \phi(\omega_u^1 + \omega_v^2).$$

where

$$S(u,v) = \begin{cases} 1 & u = v \\ -1 & u \neq v. \end{cases}$$

Let us first consider the case where $\mathbf{w}$ corresponds to a negative readout weight $a^-$. From above, we know that $\mathbf{w}$ is antiparallel. Hence, when encountering a same example, $\omega_u + \omega_u \approx 0$. If the magnitude of all coordinates are roughly equal, when encountering a different example, $\omega_u + \omega_v > 0$ about $1/4$ of the time. Comparing Eq (C.11) to Eq (C.16), we see that the expected drift for antiparallel weight vectors is roughly twice that of parallel weight vectors over long timescales. Altogether, since the number of *same* and *different* examples is balanced, we have overall that $|\Delta a^-| \propto 2 \cdot \frac{1}{4} = \frac{1}{2}$ for $a^- < 0$.

Now consider the case where $\mathbf{w}$ corresponds to a positive readout weight $a^+$. From above, we know that $\mathbf{w}$ is parallel. For large batch sizes $N$, the expected drift of a walker at initialization is $0$[7], so we expect $n^+ \approx n^-$. If the magnitude of all coordinates are roughly equal and $L$ is large, when encountering a same example, $\omega_u + \omega_v > 0$ about $1/2$ of the time. When encountering a different example, $\omega_u + \omega_v > 0$ about $1/4$ of the time. Altogether, we would therefore expected $|\Delta a^+| \propto \frac{1}{2} - \frac{1}{4} = \frac{1}{4}$.

From this rough estimate, we find that $\frac{|\Delta a^-|}{|\Delta a^+|} \approx 2$ for average negative and positive readouts. Since the rate of increase for negative readouts tends to be larger than that of positive readouts, we would expect the magnitude of negative readouts to

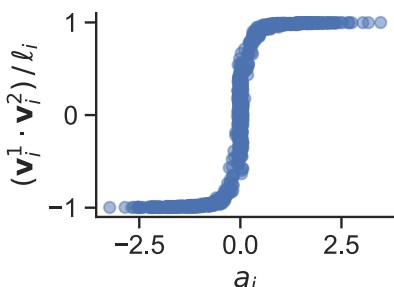

Figure C2: **Rich-regime weight structure when $L = 2$.** The model continues to develop parallel/antiparallel weights, though the magnitude of negative readouts is now about the same as the magnitude of positive readouts.

be similarly larger. In fact, if readouts start with small initialization, we may conjecture that $\frac{|\overline{a^-}|}{|\overline{a^+}|} \approx 2$. In practice, this quantity turns out to be about $1.56 \pm 0.09$ (with 2 standard deviations), computed from 10 runs [8].

Note in the case that $L$ is small (for instance, $L = 2$), our estimate $|\Delta a^+| \propto \frac{1}{4}$ breaks down since there may be only a single set of positive coordinates and no penalty is incurred on negative examples. In this case, we would have $|\Delta a^+| \propto \frac{1}{2}$, so $\frac{|\Delta a^-|}{|\Delta a^+|} = 1$. Indeed, this is exactly what we observe in the case where $L = 2$. Computing this quantity empirically yields $0.99 \pm 0.07$ (with 2 standard deviations), computed from 10 runs [9]. This outcome seems to be part of the reason why the model does not generalize well on the SD task with only $2$ symbols, despite developing parallel/antiparallel weights (Figure C2). For $L \geq 3$, there seems to be sufficient pairs of positive coordinates in parallel weight vectors to restore the situation where $|\Delta a^-| > |\Delta a^+|$.

## C.5 Test accuracy prediction

We apply our knowledge on the structure of $\mathbf{w}$ and $a$ to estimate the test accuracy of the rich regime model. Our derivation is heuristic, but seeks to capture broad phenomena rather than achieve exact precision. We validate our predicted test accuracy in Figure C3, demonstrating excellent agreement.

Recall from Section 3.1 that a model achieving the handcrafted solution exhibits perfect classification of *same* examples. Any errors are therefore accumulated from misclassifying *different* examples. The crux of our estimate stems from approximating the classification accuracy of *different* examples.

Define $I^+$ to be the set of weight indices $i$ such that $a_i > 0$, and define $I^-$ to be the set of weight indices $j$ where $a_j < 0$. Let $\mathbf{x}$ be a *different* example. Dropping constants that do not

---

[7]There is a 1/3 chance of observing a *same* example, which increments by 2. There is a 2/3 chance of observing a *different* example, which decrements by 1. Hence, the expected drift at initialization must be zero overall.

[8]The MLP has width $m = 1024$, and inputs have dimension $d = 512$. There are $L = 32$ training symbols

[9]As before, the MLP has width $m = 1024$, and inputs have dimension $d = 512$. There are $L = 2$ training symbols

affect the outcome of a classification, our model becomes

$$f(\mathbf{x}) = \sum_{i \in I^+} |a_i|\,\phi(\mathbf{w}_i \cdot \mathbf{x}) - \sum_{j \in I^-} |a_j|\,\phi(\mathbf{w}_j \cdot \mathbf{x})\,.$$

Define the weighted sums

$$\overline{a^+}_* = \frac{\sum_{i \in I^+} |a_i|\,\phi(\mathbf{w}_i \cdot \mathbf{x})}{\sum_{i \in I^+} \phi(\mathbf{w}_i \cdot \mathbf{x})}\,,$$

$$\overline{a^-}_* = \frac{\sum_{j \in I^-} |a_j|\,\phi(\mathbf{w}_j \cdot \mathbf{x})}{\sum_{i \in I^+} \phi(\mathbf{w}_j \cdot \mathbf{x})}\,.$$

Then

$$f(\mathbf{x}) = \overline{a}^+_* \sum_{i \in I^+} \phi(\mathbf{w}_i \cdot \mathbf{x}) - \overline{a}^-_* \sum_{j \in I^-} \phi(\mathbf{w}_j \cdot \mathbf{x})\,.$$

If the magnitudes of $\phi(\mathbf{w}_i \cdot \mathbf{x})$ are the same for all $i$ and all $\mathbf{x}$, then $\overline{a^+}_* = \overline{a^+} = \frac{1}{|I^+|}\sum_{i \in I^+} a_i$. Since $\mathbf{x}$ is an unseen *different* example, by symmetry we conclude that $\overline{a^+}_* = \overline{a^+}$ is a reasonable approximation. The same applies for $\overline{a^-}_* = \overline{a^-}$.

Since the magnitude of $f(\mathbf{x})$ does not affect its classification, we divide through by $\overline{a^+}$ to redefine our model as

$$f(\mathbf{x}) = \sum_{i \in I^+} \phi(\mathbf{w}_i \cdot \mathbf{x}) - \rho \sum_{j \in I^-} \phi(\mathbf{w}_j \cdot \mathbf{x})\,,$$

where $\rho = \frac{\overline{|a^-|}}{\overline{|a^+|}}$. To calculate the probability of classifying an unseen *different* example, we would like to estimate $p(f(\mathbf{x}) < 0)$, for $\mathbf{x} = (\mathbf{z}, \mathbf{z}')$ and $\mathbf{z}, \mathbf{z}' \sim \mathcal{N}(\mathbf{0}, \mathbf{I}/d)$.

Then from Eq C.3

$$\phi(\mathbf{w}_i \cdot \mathbf{x}) = \phi\left(\sum_{k=1}^{L} \left[\omega^1_{i,k}\,\mathbf{z}_k \cdot \mathbf{z} \pm \omega^2_{i,k}\,\mathbf{z}_k \cdot \mathbf{z}'\right]\right)\,.$$

Over the distribution of an unseen symbol $\mathbf{z}$

$$\mathbf{z}_k \cdot \mathbf{z} \overset{d}{=} -\mathbf{z}_k \cdot \mathbf{z}\,,$$

so we replace $\pm$ with simply $+$ in the summation. Summing across all weight vectors corresponding to the *same* class yields

$$\sum_{i \in I^+} \phi(\mathbf{w}_i \cdot \mathbf{x}) = \sum_{i \in I^+} \phi\left(\sum_{k=1}^{L} \left[\omega^1_{i,k}\,\mathbf{z}_k \cdot \mathbf{z} + \omega^2_{i,k}\,\mathbf{z}_k \cdot \mathbf{z}'\right]\right)$$

$$= \sum_{\mathbf{w}_i \cdot \mathbf{x} > 0} \sum_{k=1}^{L} \left[\omega^1_{i,k}\,\mathbf{z}_k \cdot \mathbf{z} + \omega^2_{i,k}\,\mathbf{z}_k \cdot \mathbf{z}'\right]\,.$$

Since $\mathbf{w}_i \cdot \mathbf{x} > 0$, it is likely that $\omega^1_{i,k}\,\mathbf{z}_k \cdot \mathbf{z} > 0$. Since $\mathbf{z}_k \cdot \mathbf{z}$ is approximately Normal at high $d$, we approximate the term $c^1_{i,k} \equiv \omega^1_{i,k}\,\mathbf{z}_k \cdot \mathbf{z}$ as a Half-Normal random variable. We therefore focus on characterizing the distribution of a sum over Half-Normal random variables, which we denote by $\overline{c^+}$:

$$\sum_{i \in I^+} \phi(\mathbf{w}_i \cdot \mathbf{x}) \overset{d}{=} \overline{c^+} \equiv \sum_{i=1}^{|I^+|} \sum_{k=1}^{L} \left[c^1_{i,k} + c^2_{i,k}\right]\,.$$

The individual $c^1_{i,k}$ and $c^2_{i,k}$ are learned from a finite set of training symbols, so they cannot be independently distributed. However, since training symbols *are* sampled independently, we would expect for any particular weight index $i = i_0$, the corresponding $c^1_{i_0,k}$ and $c^2_{i_0,k}$ are indeed independent. Hence, we have at least $2L$ independent terms for a particular index $i = i_0$.

The dependency structure across weight indices $i$ is more subtle, but we make a reasonable guess at their structure and later validate this heuristic with numerics. While our analysis in Appendix C assumed each weight vector evolves independently, in a training model they evolve based on the same set of inputs. As a result, significant correlations emerge across weight vectors. To understand these correlations, let us fix our training symbol index to $k = k_0$ and consider all terms $c^1_{i,k_0} = \omega^1_{i,k_0}\mathbf{z}_{k_0} \cdot \mathbf{z}$. Since they all share $\mathbf{z}_{k_0} \cdot \mathbf{z}$, these quantities are not strictly independent. However, after $T$ training steps, we have that $\omega^1_{i,k_0} = O(T)$ and $c^1_{i,k_0} = O(T)$ while $\mathbf{z}_{k_0} \cdot \mathbf{z} = O(1)$, so for our approximation we will consider the dependency incurred from $\mathbf{z}_{k_0} \cdot \mathbf{z}$ as negligible.

Let us therefore turn our attention to the coordinates $\omega^1_{i,k_0}$. If the model only ever received *same* inputs $(\mathbf{z}_{k_0}, \mathbf{z}_{k_0})$, all $\omega^1_{i,k_0}$ would be identical or zero across indices $i$. However, if we allow the model to witness *different* inputs $(\mathbf{z}_{k_0}, \mathbf{z}_\ell)$ for some $\ell \neq k_0$, we would expect a distribution of $\omega^1_{i,k_0}$ driven by the underlying initialization of $\mathbf{w}_i$ and the number of symbols $L$. Coordinates where $\omega^1_{i,k_0}(0) + \omega^1_{i,1}(0) > 0$ and $\omega^1_{i,k_0}(0) + \omega^1_{i,2}(0) < 0$ would evolve differently from coordinates where $\omega^1_{i,k_0}(0) + \omega^1_{i,1}(0) < 0$ and $\omega^1_{i,k_0}(0) + \omega^2_{i,1}(0) > 0$. If the number of training symbols increases, we would expect the number of independent coordinates $\omega^1_{i,k_0}$ to also increase. Given $L$ training symbols, we might therefore guess that the number of independent coordinates to be proportional to $L - 1$, for $L - 1$ symbols where $\ell \neq k_0$. However, we also need to account for the sign of $\mathbf{z}_\ell \cdot \mathbf{z}'$. If this quantity is positive and the corresponding readout is positive, then correlations resulting from the symbol $\mathbf{z}_\ell$ would be unimportant since they lower the probability that $\mathbf{w}_i \cdot \mathbf{x} > 0$, filtering them from the sum. (The reverse is true for weights corresponding to negative readouts.) Hence, roughly half the $L - 1$ symbols contribute to unique coordinates. The total number of unique coordinates $\omega^1_{i,k_0}$ is therefore approximately $\frac{1}{2}(L - 1)$.

If each of our $\frac{1}{2}(L - 1)$ independent coordinates carries $2L$ independent dot-product terms, we have

$$\overline{c^+} \overset{d}{=} \sum_{\ell=1}^{L(L-1)} c_\ell\,,$$

where $c_\ell$ is distributed Half-Normal with mean $0$ and some variance $\sigma^2$, which will cancel in the final calculation.

Applying the central limit theorem together with the first and second moments of a Half-Normal distribution reveals that

$$\overline{c^+} \sim \mathcal{N}\left((L^2 - L)\sigma\sqrt{\frac{2}{\pi}},\ (L^2 - L)\left(1 - \frac{2}{\pi}\right)\right)\,.$$

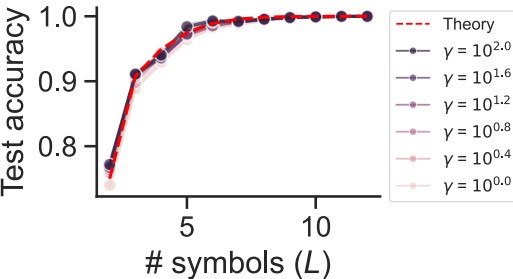

Figure C3: **Rich-regime test accuracy.** We demonstrate close agreement between the theoretically predicted and empirically measured rich-regime test accuracy. The rich-regime parametrization explored in the main text corresponds to $\gamma = 1$. To confirm that our results hold for arbitrarily rich models, we also plot accuracies attained in the *ultra-rich* regime $\gamma \gg 1$ (Atanasov et al., 2024). In all cases, our predictions continue to hold.

As we noted before, $\mathbf{z}_k \cdot \mathbf{z} \stackrel{d}{=} \mathbf{z}_k \mathbf{z}'$ for large $d$, so the distribution of $\sum \phi(\mathbf{w} \cdot \mathbf{x})$ is the same regardless if the weight vectors $\mathbf{w}$ are parallel or antiparallel. Hence, $\overline{c^+} \stackrel{d}{=} \overline{c^-}$. The distribution of our output is therefore

$$f(\mathbf{x}) \stackrel{d}{=} \overline{c^+} - \rho\,\overline{c^-}.$$

Our final probability of classifying an unseen *different* example correctly is

$$p(f(\mathbf{x}) < 0) = \Phi\left( \sqrt{\frac{(2L^2 - 2L)(\rho - 1)^2}{(\pi - 2)(\rho^2 + 1)}} \right), \qquad \text{(C.18)}$$

where $\Phi$ is the CDF of a standard Normal.

From Section C.4.3, we found that $\rho \approx 1.5$ for $L \geq 3$ and $\rho = 1$ for $L = 2$. Plugging this value into Eq (C.18) allows us to compute the probability of classifying an unseen *different* example. If the model classifies all unseen *same* examples correctly, the total test accuracy of the rich regime model is given by $\frac{1}{2} + \frac{1}{2} p(f(\mathbf{x}) < 0)$, yielding the expression we report in Eq (3). We validate this prediction in Figure C3, showing excellent agreement with the measured test accuracy of a rich model.

Two important details to note:

- The test accuracy of the rich model rises rapidly with $L$. By $L = 3$, the model already attains over 90 percent test accuracy.

- The test error does not depend on the input dimension $d$. The impact of $d$ is captured in the variance of $\sigma^2$ of our Half-Normal random variables $c$, which cancels in the final calculation.

In this way, we see how the conceptual parallel/antiparallel representations of the same-different model lead to highly efficient learning and insensitivity to input dimension, completing our analysis of the rich regime.

## D    Lazy regime details

In the lazy regime, we will demonstrate that the model requires 1) far more training symbols than in the rich regime to learn the SD task, and 2) the model's test accuracy depends explicitly on the input dimension $d$.

A lazy MLP's learning dynamics can be described using kernel methods. In particular, the case where $\gamma \to 0$ corresponds to using the Neural Tangent Kernel (NTK) (Jacot et al., 2018), in which weights evolve linearly around their initialization. We demonstrate that the number of training symbols required to generalize using the NTK grows quadratically with the input dimension.

Recall that our model has form

$$f(\mathbf{x}) = \sum_{i=1}^{m} a_i \phi(\mathbf{w}_i \cdot \mathbf{x}).$$

If there are $P$ unique training examples in our dataset, we may rewrite our model in its dual form

$$f(\mathbf{x}) = \sum_{j=1}^{P} b_j K(\mathbf{x}, \mathbf{x}_j), \qquad \text{(D.1)}$$

for the kernel

$$K(\mathbf{x}, \mathbf{x}_j) = \frac{1}{m} \sum_{i=1}^{m} \phi(\mathbf{w}_i \cdot \mathbf{x})\, \phi(\mathbf{w}_i \cdot \mathbf{x}_j).$$

For ease of exposition, we assume that inputs $\mathbf{x}$ lie on the unit sphere $\mathbf{x} \in \mathbb{S}^{2d-1}$. This is exactly true (up to a constant radius) as $d \to \infty$. For width $m \to \infty$ and ReLU activations $\phi$, the analytic form of the NTK kernel $K$ is known to be

$$K(u) = u\left( 1 - \frac{1}{\pi}\cos^{-1}(u) \right) + \frac{1}{2\pi}\sqrt{1 - u^2}.$$

where $u = \mathbf{x} \cdot \mathbf{x}'$ (Cho & Saul, 2009).

With the setup complete, we present our central result.

**Theorem 2.** *Let $f$ be an infinite-width ReLU MLP as given in Eq D.1, with an NTK kernel. Suppose inputs $\mathbf{x}$ are restricted to lie on the unit sphere $\mathbf{x} \in \mathbb{S}^{2d-1}$. If $f$ is trained on a dataset consisting of $P$ points constructed from $L$ symbols with input dimension $d$, then the test error of $f$ is upper bounded by $O\left( \exp\left\{ -\frac{L}{d^2} \right\} \right)$.*

*Proof.* Our proof strategy proceeds as follows. We restrict the space over which our dual coefficients $b$ can vary to a convenient subset, and upper bound the achievable test error of $f$ over this restricted parameter space. Because the restricted parameter space is a subset of the full parameter space, our derived upper bound applies to the unrestricted model as well.

We restrict the dual coefficients $b$ as follows. Let $I^+$ be the set of all indices $i$ such that $\mathbf{x}_i$ is a *same* example, and $I^-$ be the set of all indices $j$ such that $\mathbf{x}_j$ is *different*. Then for all $i \in I^+$, we fix $b_i = b^+ > 0$. For all $j \in I^-$, we fix $b_j = b^- < 0$. Hence, we effectively tune just two parameters: $b^+$ and $b^-$.

We also set a number of coefficients $b$ to zero. Given a dataset with symbols $z_1, z_2, \ldots, z_L$, partition the symbols such that set $\mathcal{S}_1 = \{z_1, z_2, \ldots z_{L/3}\}$ and $\mathcal{S}_2 = \{z_{L/3+1}, z_{L/3+2}, \ldots z_L\}$. Consider the kernel coefficient $b_k$, which corresponds to a training example $x_k = (z_{\ell_1}; z_{\ell_2})$. If $z_{\ell_1} = z_{\ell_2}$ and $z_{\ell_1} \notin \mathcal{S}_1$, then we fix $b_k = 0$. If $z_{\ell_1} \neq z_{\ell_2}$, then we check three conditions: (1) $z_{\ell_1} \in \mathcal{S}_2$, (2) $\ell_2 - \ell_1 = 1$, and $\ell_1$ is odd. If any one of these conditions is violated, we set $b_k = 0$.

This procedure for deciding whether $b_k = 0$ ensures that the remaining nonzero terms in Eq (D.1) are independent, and that there are an equal number of *same* and *different* examples remaining. The set $\mathcal{S}_1$ determines the symbols that contribute to *same* examples. The disjoint set $\mathcal{S}_2$ determines the symbols that contribute to *different* examples. We further stipulate that *different* examples do not contain overlapping symbols, leading to the three conditions enumerated above. Note, to construct a dataset such that there are $P$ nonzero terms in our kernel sum, we require $L \propto P$ symbols.

First, suppose $x$ is a *same* test example. Since we restricted the summands to be independent in our kernel function, the probability of mis-classifying $x$ can bounded through a straightforward application of Hoeffding's inequality

$$p(f(x) < 0) \leq \exp\left\{ -\frac{2\mathbb{E}[f(x)]^2}{Pc^2} \right\} \qquad \text{(D.2)}$$

where $P$ is the size of the training set and $c$ is a constant related to the range of individual summands $b_j K(x, x_j)$. Note, $b_j$ can be arbitrarily small without changing the classification and $0 \leq K(u) < 3$, so $c$ is finite. Distributing the expectation, we have

$$\mathbb{E}[f(x)] = \sum_{i \in I^+} b^+ \mathbb{E}[K(x, x_i)] + \sum_{j \in I^-} b^- \mathbb{E}[K(x, x_j)]. \qquad \text{(D.3)}$$

Taylor expanding $K$ to second order in $u$ reveals that

$$\mathbb{E}[K(u)] = \frac{1}{2\pi} + \frac{\mathbb{E}[u]}{2} + \frac{3\mathbb{E}[u^2]}{4\pi} + o(\mathbb{E}[u^2]) \qquad \text{(D.4)}$$

Since input symbols are normally distributed with mean zero, we know that $\mathbb{E}[u] = 0$ and $\mathbb{E}[u^2] \propto 1/d$. Furthermore, if $x_j$ is a *same* training example and $x_k$ is a *different* training example, inspecting second moments reveals that $\mathbb{E}[(x \cdot x_j)^2] = 2\mathbb{E}[(x \cdot x_k)^2]$, for an unseen *same* example $x$. Thus, provided that $|b^-/b^+| < 2$, substituting (D.4) into (D.3) yields

$$\mathbb{E}[f(x)] = O\left(\frac{P}{d}\right),$$

which implies that

$$p(f(x) < 0) \leq O\left(\exp\left\{ -\frac{P}{d^2} \right\}\right).$$

Now suppose $x$ is a *different* test example. If $x^+$ is a *same* training example and $x^-$ is a *different* training example, then the first and second moments of $x \cdot x^+$ are equal to that of

$x \cdot x^-$. Hence, (D.4) and (D.3) suggest that if $|b^-| - |b^+| = O(1)$, then $\mathbb{E}[f(x)] = -O(P)$. Applying Hoeffding's a second time suggests that

$$p(f(x) > 0) = O(\exp\{-P\}).$$

Note, it is possible to satisfy both $|b^-| - |b^+|$ and $|b^-/b^+| < 2$, for example with $b^+ = 1$ and $b^- = 1.1$.

The test error overall is dominated by the contribution from mis-classifying *same* examples $p(f(x) < 0) = O(\exp\{-P/d^2\})$. Because of our independence restriction on the dual coefficients $b$, in order to produce $P$ training examples, we require $L \propto P$ training symbols. The test error of the model overall is therefore upper bounded by $O\left(\exp\left\{ -\frac{L}{d^2} \right\}\right)$. $\qquad \square$

Hence, in order to maintain a constant error rate, our bound suggests that the number of training symbols $L$ should scale as $L \propto d^2$. While this scaling is an upper bound on the true error rate of a lazy model, Figure 1f suggests that this quadratic relationship remains descriptive of the full model. There are two important consequences of this result:

1. For a large $d$, the lazy model requires substantially more training symbols to learn the SD task than the rich model. In Appendix D, we found that the rich model can generalize with as few as $L = 3$ symbols. In contrast, Figure 1f suggests the lazy model will often require hundreds or thousands of training symbols to generalize.

2. For a fixed number of training symbols, a lazy model's performance decays as $d$ increases. Unlike in the rich case, there is an explicit dependency on $d$ in the test error for the lazy model, hurting its performance as $d$ grows larger.

In this way, we see how a lazy model can leverage the differing statistics of *same* and *different* examples to accomplish the SD task, but at the cost of exhaustive training data and strong sensitivity to input dimension.

## E Bayesian posterior calculations

In Section 3.4, we compute with the posteriors corresponding to two different idealized models: one that generalizes to novel symbols based on the true underlying symbol distribution, and one that memorizes the training symbols. Below, we present the Bayes optimal classifier for our noisy same different, and derive the posteriors associated with these two settings.

## E.1 Generalizing prior

We define the following data generating process that constitutes a prior which generalizes to arbitrary, unseen symbols.

$$r \sim \text{Bernoulli}\left(p = \frac{1}{2}\right)$$

$$\mathbf{s}_1 \sim \mathcal{N}\left(0, \frac{1}{d}\right)$$

$$\mathbf{s}_2 \sim \begin{cases} \delta(\mathbf{s}_1) & r = 1 \\ \mathcal{N}\left(0, \frac{1}{d}\right) & r = 0 \end{cases}$$

$$\mathbf{z}_1 \sim \mathcal{N}\left(\mathbf{s}_1, \frac{\sigma^2}{d}\right)$$

$$\mathbf{z}_2 \sim \mathcal{N}\left(\mathbf{s}_2, \frac{\sigma^2}{d}\right)$$

The quantity $r$ represents either a *same* or *different* relation. Variables $\mathbf{s}_1, \mathbf{s}_2$ are symbols matching their description in Section 2. The notation $\delta(\mathbf{s}_1)$ denotes a Delta distribution centered at $\mathbf{s}_1$. Hence, $\mathbf{s}_1 = \mathbf{s}_2$ if $r = 1$, and differ otherwise. Typically, we consider the noiseless case $\sigma^2 = 0$, but to develop a Bayesian treatment, we allow $\sigma^2 > 0$. We approximate the noiseless case by considering $\sigma^2 \to 0$.

The Bayes optimal classifier is

$$\hat{y}_{bayes} = \begin{cases} 1 & p(r = 1 \,|\, \mathbf{z}_1, \mathbf{z}_2) \geq \frac{1}{2} \\ 0 & \text{otherwise} \end{cases} \tag{E.1}$$

From Bayes rule, we know that

$$p(r \,|\, \mathbf{z}_1, \mathbf{z}_2) \propto p(\mathbf{z}_1, \mathbf{z}_2 \,|\, r)\, p(r).$$

Since $r$ is sampled with equal probability $1$ or $0$, we have simply

$$p(r \,|\, \mathbf{z}_1, \mathbf{z}_2) \propto p(\mathbf{z}_1, \mathbf{z}_2 \,|\, r).$$

We use the notation $\mathcal{N}(\mathbf{x}; \mu, \sigma^2)$ to mean the PDF of a Normal distribution evaluated at $\mathbf{x}$, with mean $\mu$ and covariance $\sigma^2 \mathbf{I}$. We then compute

$$p(\mathbf{z}_1, \mathbf{z}_2 \,|\, r = 1) = \int \mathcal{N}\left(\mathbf{z}_1; \mathbf{s}, \frac{\sigma^2}{d}\right) \mathcal{N}\left(\mathbf{z}_2; \mathbf{s}, \frac{\sigma^2}{d}\right) \mathcal{N}\left(\mathbf{s}; \mathbf{0}, \frac{\sigma^2}{d}\right) d\mathbf{s}$$

$$= \left(\frac{d}{2\pi\sqrt{\sigma^2(2 + \sigma^2)}}\right)^d \exp\left\{-\frac{d}{2\sigma^2}\left(\frac{1 + \sigma^2}{2 + \sigma^2}\left(||\mathbf{z}_1||^2 + ||\mathbf{z}_2||^2\right) - \frac{2}{2 + \sigma^2}\left(\mathbf{z}_1 \cdot \mathbf{z}_2\right)\right)\right\}, \tag{E.2}$$

$$p(\mathbf{z}_1, \mathbf{z}_2 \,|\, r = 0) = \int \int \mathcal{N}\left(\mathbf{z}_1; \mathbf{s}_1, \frac{\sigma^2}{d}\right) \mathcal{N}\left(\mathbf{z}_2; \mathbf{s}_2, \frac{\sigma^2}{d}\right) \mathcal{N}\left(\mathbf{s}_1; \mathbf{0}, \frac{\sigma^2}{d}\right) \mathcal{N}\left(\mathbf{s}_2; \mathbf{0}, \frac{\sigma^2}{d}\right) d\mathbf{s}_1\, d\mathbf{s}_2$$

$$= \left(\frac{d}{2\pi(1 + \sigma^2)}\right)^d \exp\left\{-\frac{d}{2}\left(\frac{1}{1 + \sigma^2}\right)\left(||\mathbf{z}_1||^2 + ||\mathbf{z}_2||^2\right)\right\}. \tag{E.3}$$

Using Eq (E.2) and (E.3), we compute

$$p(r = 1 \,|\, \mathbf{z}_1, \mathbf{z}_2) = \frac{p(\mathbf{z}_1, \mathbf{z}_2 \,|\, r = 1)}{p(\mathbf{z}_1, \mathbf{z}_2 \,|\, r = 1) + p(\mathbf{z}_1, \mathbf{z}_2 \,|\, r = 0)},$$

which we plug back into Eq (E.1) to obtain our Bayes classifier under a generalizing prior.

## E.2 Memorizing prior

The data generating process for a model that memorizes the training data is similar to the generalizing model, but the crucial difference is that the symbols $\mathbf{s}$ are now distributed uniformly across the training symbols rather than sampled from their population distribution.

Let $\hat{\mathbf{s}}_1, \hat{\mathbf{s}}_2, \ldots, \hat{\mathbf{s}}_L$ be the set of $L$ training symbols. Then the data generating process is given by

$$r \sim \text{Bernoulli}\left(p = \frac{1}{2}\right)$$

$$\mathbf{s}_1 \sim \text{Uniform}\{\hat{\mathbf{s}}_1, \hat{\mathbf{s}}_2, \ldots, \hat{\mathbf{s}}_L\}$$

$$\mathbf{s}_2 \sim \begin{cases} \delta(\mathbf{s}_1) & r = 1 \\ \text{Uniform}\{\hat{\mathbf{s}}_1, \hat{\mathbf{s}}_2, \ldots, \hat{\mathbf{s}}_L\} \setminus \{\mathbf{s}_1\} & r = 0 \end{cases}$$

$$\mathbf{z}_1 \sim \mathcal{N}\left(\mathbf{s}_1, \frac{\sigma^2}{d}\right)$$

$$\mathbf{z}_2 \sim \mathcal{N}\left(\mathbf{s}_2, \frac{\sigma^2}{d}\right)$$

As before, we compute the probabilities $p(\mathbf{z}_1, \mathbf{z}_2 \,|\, r = 1)$ and $p(\mathbf{z}_1, \mathbf{z}_2, \,|\, r = 0)$, which are given by

$$p(\mathbf{z}_1, \mathbf{z}_2 \mid r = 1) = \frac{1}{L} \sum_{i=1}^{L} p(\mathbf{z}_1, \mathbf{z}_2 \mid \hat{\mathbf{s}}_i)$$

$$= \frac{1}{L} \sum_{i=1}^{L} \mathcal{N}\left(\mathbf{z}_1; \hat{\mathbf{s}}_i, \frac{\sigma^2}{d}\right) \mathcal{N}\left(\mathbf{z}_2; \hat{\mathbf{s}}_i, \frac{\sigma^2}{d}\right)$$

$$= \left(\frac{d}{2\pi\sigma^2}\right)^d \exp\left\{-\frac{d}{2\sigma^2}\left(\frac{1}{2}\left(||\mathbf{z}_1||^2 + ||\mathbf{z}_2||\right)^2 - \mathbf{z}_1 \cdot \mathbf{z}_2\right)\right\} \left(\frac{1}{L} \sum_{i=1}^{L} \exp\left\{-\frac{d}{\sigma^2}\left|\left|\hat{\mathbf{s}}_i - \frac{\mathbf{z}_1 + \mathbf{z}_2}{2}\right|\right|^2\right\}\right), \quad \text{(E.4)}$$

$$p(\mathbf{z}_1, \mathbf{z}_2 \mid r = 0) = \frac{1}{L(L-1)} \sum_{i \neq j} p(\mathbf{z}_1 \mid \hat{\mathbf{s}}_i) \, p(\mathbf{z}_2 \mid \hat{\mathbf{s}}_j)$$

$$= \frac{1}{L(L-1)} \sum_{i \neq j}^{L} \mathcal{N}\left(\mathbf{z}_1; \hat{\mathbf{s}}_i, \frac{\sigma^2}{d}\right) \mathcal{N}\left(\mathbf{z}_2; \hat{\mathbf{s}}_j, \frac{\sigma^2}{d}\right)$$

$$= \frac{1}{L(L-1)} \sum_{i \neq j} \left(\frac{d}{2\pi\sigma^2}\right)^d \exp\left\{-\frac{d}{2\sigma^2}||\mathbf{z}_1 - \hat{\mathbf{s}}_i||^2\right\} \exp\left\{-\frac{d}{2\sigma^2}||\mathbf{z}_2 - \hat{\mathbf{s}}_j||^2\right\}. \quad \text{(E.5)}$$

Using Eq (E.4) and (E.5), we compute Eq (E.1) to obtain our Bayes classifier under a memorizing prior.

## F  Rich and lazy scaling

We review rich and lazy regime scaling in our setting. In particular, we consider learning dynamics as we increase the input dimension $d$ (Saad & Solla, 1995; Biehl & Schwarze, 1995; Goldt et al., 2019). This setting differs from other rich-regime studies, where scaling is considered with respect to increasing width $m$. In particular, *maximal update* ($\mu$P) and the related mean-field parameterizations consider an infinite-width limit (Yang & Hu, 2021; Mei et al., 2018; Rotskoff & Vanden-Eijnden, 2022). Our analysis holds $m$ fixed.

Recall that our model is given by

$$f(\mathbf{x}; \theta) = \frac{1}{\gamma\sqrt{d}} \sum_{i=1}^{m} a_i \phi(\mathbf{w}_i \cdot \mathbf{x}).$$

Let $\theta(t)$ be the value of the parameters $\theta$ at time-step $t$. Crucially, to permit a valid interpolation between rich and lazy learning regimes, our MLP is *centered*: $f(\mathbf{x}; \theta(0)) = 0$. Following Chizat et al. (2019), we enforce centering by subtracting the initial logit from every prediction. Hence, our classifier takes the form

$$\tilde{f}(\mathbf{x}; \theta) = f(\mathbf{x}; \theta) - f(\mathbf{x}; \theta(0)).$$

We use $\tilde{f}$ as our centered MLP in all experiments.

To see how changing $\gamma$ interpolates between rich and lazy learning regimes, recall that learning richness is a description of activation change over the course of training. One way to operationalize this description is to define *rich* learning as the case in which parameters $\theta$ change substantially in comparison with changes in the model output $\tilde{f}$, and *lazy* learning

as the case in which $\theta$ change very little with respect to the model output.

Consider the change in $\tilde{f}$ after one step of gradient descent. For a learning rate $\alpha$ and training set size $P$, we update our parameters as

$$\theta(1) = \theta(0) - \alpha \nabla_\theta \left(\frac{1}{P} \sum_{p=1}^{P} \mathcal{L}(y_p, \tilde{f}(\mathbf{x}_p; \theta))\right)$$

$$= \theta(0) - \frac{\alpha}{P} \sum_{p=1}^{P} (y_p - \sigma(\tilde{f}(\mathbf{x}_p; \theta))) \nabla_\theta \tilde{f}(\mathbf{x}_p; \theta).$$

Note that for an input $\mathbf{x}$,

$$\frac{\partial \tilde{f}}{\partial a_i} = \frac{1}{\gamma\sqrt{d}} \phi(\mathbf{w}_i \cdot \mathbf{x}),$$

$$\frac{\partial \tilde{f}}{\partial \mathbf{w}_i} = \frac{1}{\gamma\sqrt{d}} a_i \phi'(\mathbf{w}_i \cdot \mathbf{x}) \mathbf{x}.$$

Define

$$\Delta a_i \equiv \frac{1}{P} \sum_{p=1}^{P} (y_p - \sigma(\tilde{f}(\mathbf{x}_p; \theta))) \frac{\partial \tilde{f}}{\partial a_i},$$

$$\Delta \mathbf{w}_i \equiv \frac{1}{P} \sum_{p=1}^{P} (y_p - \sigma(\tilde{f}(\mathbf{x}_p; \theta))) \frac{\partial \tilde{f}}{\partial \mathbf{w}_i}.$$

Substituting our weight updates into our model reveals that

$$\tilde{f}(\mathbf{x}; \theta(1)) = \frac{\alpha}{\gamma^2 d} \sum_{i=1}^{m} \Big[ \Delta a_i \phi(\Delta w_i \cdot \mathbf{x}) / (\gamma\sqrt{d})$$
$$+ \Delta a_i \phi(\mathbf{w}(0) \cdot \mathbf{x})$$
$$+ a_i(0) \phi(\Delta \mathbf{w}_i \cdot \mathbf{x}) \Big]. \quad \text{(F.1)}$$

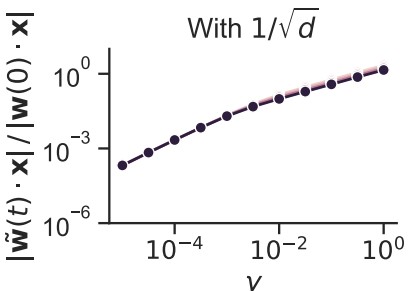
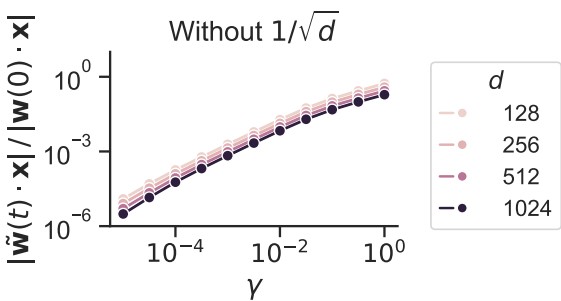

Figure F1: **Activation scales with $\gamma$ and $d$.** We plot the average absolute activation change across 6000 test examples as a function of $\gamma$ (for $m = 4096$), normalized by the initial activation size $|\mathbf{w}(0) \cdot \mathbf{x}|$. Higher $\gamma$ leads to more activation change. In the absence of a $1/\sqrt{d}$ prefactor, the activation change scales inversely with $d$. Including the $1/\sqrt{d}$ prefactor suppresses this change.

Observe that $|\Delta a_i \phi(\mathbf{w}(0) \cdot \mathbf{x})| = O_d(1)$ and $|a_i(0)\phi(\Delta\mathbf{w}_i \cdot \mathbf{x})| = O_d(1/\sqrt{d})$ for a test point $\mathbf{x}$. If we adopt a learning rate $\alpha = \gamma^2 d$, then we have overall

$$|\tilde{f}(\mathbf{x};\theta(1))| = O_d(1). \tag{F.2}$$

In this way, we find that the model output changes by a constant amount relative to the input dimension $d$. Meanwhile, the parameters change by a total magnitude

$$||\theta(1) - \theta(0)|| = O_d(\alpha(|\Delta a_i| + ||\Delta\mathbf{w}_i||) = O_d(\gamma\sqrt{d}).$$

At initialization, we have that

$$||\theta(0)|| = O_d(|a_i(0)| + ||\mathbf{w}_i(0)||) = O_d(\sqrt{d}),$$

so the change in weights relative to the scale of their initialization is simply

$$||\theta(1) - \theta(0)||/||\theta(0)|| = O_d(\gamma). \tag{F.3}$$

All together, after one gradient step, while the model output changes by a constant amount with respect to $d$, the model parameters change by $\gamma$ relative to their initialization. For $\gamma \to 0$, the initialization dominates (even as the model output changes), resulting in lazy learning. For increasing $\gamma$, the parameters move proportionally further from their initialization, resulting in progressively rich learning.

Peculiar to our setting is the additional $1/\sqrt{d}$ factor in the output scale of the MLP, not found in other rich-regime studies that consider width scaling (Yang & Hu, 2021; Mei et al., 2018; Rotskoff & Vanden-Eijnden, 2022). In the absence of the $1/\sqrt{d}$ factor, Eqs (F.1) and (F.2) suggest that we should adjust our learning rate to be $\alpha = \gamma^2$ in order to maintain a stable $O_d(1)$ output change with increasing $d$. However, the relative weight change in Eq (F.3) now becomes $O_d(\gamma/\sqrt{d})$. For fixed $\gamma$, a model becomes lazier as $d$ increases. Hence, to maintain consistent richness, we require an additional $1/\sqrt{d}$ prefactor on the MLP (along with the corresponding $\alpha = \gamma^2 d$ learning rate).

Figure F1 illustrates these conclusions. Increasing $\gamma$ increases the change in activations $|\tilde{\mathbf{w}}(t) \cdot \mathbf{x}|$ for a test example $\mathbf{x}$. In the absence of the $1/\sqrt{d}$ prefactor, increasing $d$ also decreases the change in activations.

## G Model and task details

We enumerate all model and task configurations in this Appendix. Exact details are available in our code, https://github.com/wtong98/equality-reasoning, which can be run to reproduce all plots in this manuscript.

### G.1 Model

In all experiments, we use a two-layer MLP without biases that takes inputs $\mathbf{x} \in \mathbb{R}^d$ and outputs

$$f(\mathbf{x}) = \frac{1}{\gamma\sqrt{d}} \sum_{i=1}^{m} a_i \phi(\mathbf{w}_i \cdot \mathbf{x}),$$

where $\phi$ is a point-wise ReLU nonlinearity and $\gamma$ is a hyperparamter that governs learning richness. Our MLP is centered using the procedure described in Appendix F. To produce a classification, $f$ is passed through a standard logit link function

$$\hat{y} = \frac{1}{1 + e^{-f}}.$$

Parameters are initialized based on $\mu$P (Yang et al., 2022). Specifically, we initialize our weights as

$$a_i \sim \mathcal{N}(0, 1/m),$$
$$\mathbf{w}_i \sim \mathcal{N}(\mathbf{0}, \mathbf{I}/m).$$

We train the model using stochastic gradient descent on binary cross entropy loss. Following Atanasov et al. (2024), we set the learning rate $\alpha$ as $\alpha = \gamma^2 d \alpha_0$ for $\gamma \leq 1$ and $\alpha = \gamma\sqrt{d}\alpha_0$ for $\gamma > 1$. The base learning rate $\alpha_0$ is task-specific, and varies from $0.01$ to $0.5$. To measure a model's performance, we train for a large, fixed number of iterations past convergence in training accuracy, and select the best test accuracy from the model's history.

### G.2 Same-Different

The same-different task consists of input pairs $\mathbf{z}_1, \mathbf{z}_2 \in \mathbb{R}^d$, where $\mathbf{z}_i = \mathbf{s}_i + \eta_i$. The labeling function $y$ is given by

$$y(\mathbf{z}_1, \mathbf{z}_2) = \begin{cases} 1 & \mathbf{s}_1 = \mathbf{s}_2 \\ 0 & \mathbf{s}_1 \neq \mathbf{s}_2 \end{cases}.$$

We sample these quantities as

$$\mathbf{s} \sim \mathcal{N}(\mathbf{0}, \mathbf{I}/d),$$
$$\eta \sim \mathcal{N}(\mathbf{0}, \sigma\mathbf{I}/d).$$

A training set is sampled such that half the training examples belong to class $1$, and half belong to class $0$. Crucially, the training set consists of $L$ fixed symbols $\mathbf{s}_1, \mathbf{s}_2, \ldots, \mathbf{s}_L$ sampled prior to the experiment. All training examples are constructed from these $L$ symbols. During testing, symbols are sampled afresh, forcing the model to generalize. If the noise variance $\sigma$ is not explicitly stated, then we take it to be $\sigma = 0$. We use a base learning rate $\alpha_0 = 0.1$ with batches of size 128.

### G.3  PSVRT

The PSVRT task consists of a single-channel square image with two blocks of bit-patterns. If the bit-patterns match exactly, then the image belongs to the *same* class. If the bit-patterns differ, then the image belongs to the *different* class. Images are flattened before being passed to the MLP.

Images are patch-aligned to prevent overlapping bit-patterns. An image is tiled by non-overlapping square regions which may be filled by bit-patterns. No two bit-patterns may share a single patch. Unless otherwise stated, we use patches that are 5 pixels to a side, and images that are 5 patches to a side, for a total of 25 by 25 pixels.

One important feature of PSVRT is that the inputs do not grow in norm as their dimension increases. Because there are only ever two patches in an image, regardless of its size, the total norm of the input remains constant regardless of the image dimensions. As a result, the $1/\sqrt{d}$ scaling on the MLP output is extraneous for PSVRT, and we remove it in these experiments.

A subset of all possible bit-patterns are used for training. The remaining unseen bit-patterns are used for testing. We use a base learning rate $\alpha_0 = 0.5$ with batches of size 128.

### G.4  Pentomino

The Pentomino task consists of a single-channel square image with two pentomino shapes. If the shapes are the same (up to rotation, but not reflection), then the image belongs to the *same* class. If the bit-patterns differ, then the image belongs to the *different* class. Images are flattened before being passed to the MLP.

Like before, images are patch-aligned. To provide a border around each pentomino, patches are 7 pixels to a side. Unless otherwise stated, images are 2 patches to a side, for a total of 14 by 14 pixels.

As with PSVRT, the inputs for Pentomino do not grow in norm as their dimension icnreases. There are only ever two pentomino shapes in an image, regardless of its dimension, so the total norm of the input remains constant. Like with PSVRT, we remove the $1/\sqrt{d}$ output scaling on the MLP for these experiments.

There are a total of 18 possible pentomino shapes. A subset of these 18 is held out for testing, and the model trains on the remainder. To improve training stability, mild Gaussian blurs are randomly applied to training images, but not testing images. We use a base learning rate $\alpha_0 = 0.5$ with batches of size 128.

### G.5  CIFAR-100

The CIFAR-100 same-different task consists of full-color images taken from the CIFAR-100 dataset. Images are 32 by 32 pixels, and depict 1 of among 100 different classes. To form an input example, we place two images side-by-side, forming a larger 64 by 32 pixel image. If the images come from the same class (but are not necessarily the same exact image), the example belongs to the *same* class. If the images come from different classes, the example belong to the *different* class.

To separate an MLP's ability to reason about equality from its ability to extract meaningful visual features, we first pass the image through a VGG-16 backbone pretrained on ImageNet. Activations are then taken from an intermediate layer, flattened, and passed to the MLP. Because VGG-16 activations are coordinate-wise $O(1)$ in magnitude, we normalize them by $1/\sqrt{d}$ before input to the model. The resulting performance of the MLP from activations of each layer are plotted in Figure G1.

Of the 100 total classes, a subset is held out for testing, and the model trains on the remainder. We use a base learning rate $\alpha_0 = 0.01$ with batches of size 128.

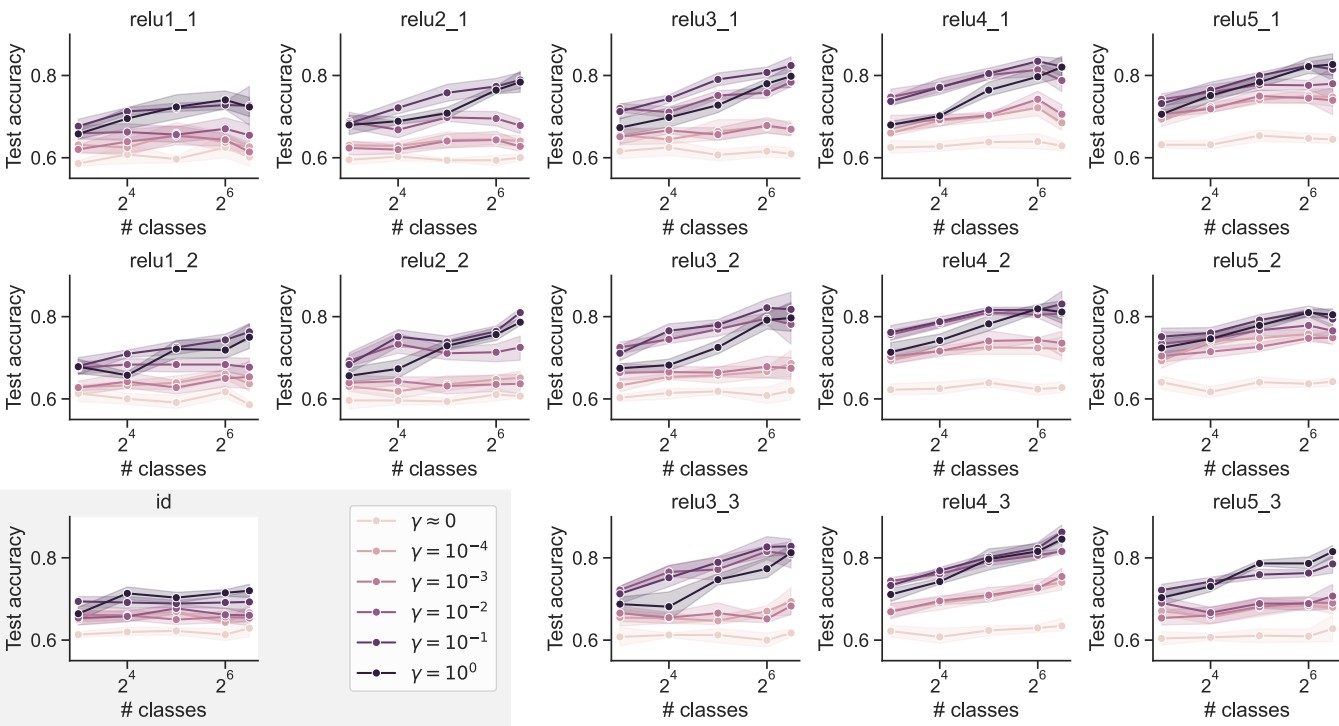

Figure G1: **CIFAR-100 same-different accuracy across different VGG-16 activations.** Activations are named by `relu[block]_[layer]` The plot with name `id` corresponds to using the raw images directly without first preprocessing in VGG-16. Earlier and later layers demonstrate an interesting collapse where learning richness does not seem to impact classification accuracy very strongly. Intermediate layers suggest that greater learning richness tends to perform better, though the richest model tends to do poorly. Shaded error regions correspond to 95 percent confidence intervals estimated from 6 runs.

