# OpenReview forum: "Learning richness modulates equality reasoning in neural networks"
_ccneuro.org/CCN/2025/Proceedings — CCN 2025 Proceedings asProceedingsPoster_

### Official Review · Reviewer_rMwz · 2025-03-28
**Review for Learning richness modulates equality reasoning in neural networks**

**Soundness:** 3
**Clarity:** 2

**Comments:**

### After author response: with the updated information, I find the paper solid enough from a technical perspective for acceptance.

Summary:

This paper studies behavior of MLPs in same-different tasks from a theoretical perspective. The authors make a key distinction between conceptual and perceptual behavior. The former is characterized by efficient learning, and insensitivity to spurious perceptual details. The latter shows strong sensitivity to spurious perceptual details, and requires exhaustive training. It is then argued that MLP behavior is controlled by their learning richness, which is verified in several simulations.

General assessment:

In general, the paper was easy to read, well-structured, and logically coherent. It was technically solid and clean, and despite being quite technical it was written in a way that can be followed by someone not too familiar with this literature (such as me). It was cool to see that the insights translated to more realistic settings in which assumptions are lifted (at least most of the time). Perhaps the biggest weakness is that it is rather narrow in scope and probably only of interest to a subset of the CCN audience.

More specific points and questions:

* The paper could benefit from more details regarding the training procedure. From context, I assume that all models are trained until convergence with plain SGD and then evaluated. Do all converged models reach 100% accuracy on the training set? How is the learning rate set? The authors mention “appropriately scaled” learning rate but this is not further specified.
* How exactly is test accuracy computed? Is it based on thresholding predictions by the model or averaging over the output probabilities produced by the model? I assume it is the former. If so, do the results change if one instead averages over output probabilities?
* Learning richness seems to be equivalent to a temperature parameter unless I am missing something. If so, what is all the fuzz about?
* The $\mu P$ parametrization is introduced and referred to later on but never really explained – I was a bit lost here.
* Is the max-margin solution from theorem 1 actually used somewhere? The authors seem to only rely on the heuristic construction for the ReLU MLP introduced afterward.

**Expertise:**

1

**Interest:**

1

---

> ### Author Rebuttal · Authors · 2025-04-14
>
> Thanks for the comments and suggestions! We’re glad you found our manuscript to be well-written and technically strong. We’ve incorporated all edits in a revised manuscript uploaded to OpenReview.
>
> Unfortunately, due to the tight word limit imposed on rebuttals, we are unable to respond to all your questions. We offer brief answers below where possible, but please do ask additional questions to offer us the opportunity to elaborate. Apologies for the inconvenience!
>
> > The paper could benefit from more details regarding the training procedure.
>
> Great suggestion. We have expanded appendix G in particular to further enumerate all experimental settings. We have more to say, so please ask a follow-up if needed
>
> > How exactly is test accuracy computed? Is it based on thresholding predictions by the model or averaging over the output probabilities produced by the model?
>
> Great question that we unfortunately lack the space to answer, please ask again
>
> > Learning richness seems to be equivalent to temperature?
>
> Learning richness describes the degree of change in activations over the course of learning. It is a distinct concept from softmax temperature, which governs the degree of uniformity in output probabilities. While both richness and temperature can be controlled by scaling the model output, richness additionally requires a corresponding scaling of initialization and learning rate. Without these two, the model may inadvertently fall into a lazy regim or fail to learn at all in finite time. The importance of richness stems in part from the surprising fact that it can be controlled by a single hyperparameter $\gamma$, when coupled correctly with learning rate and initialization scale.
>
> > The μP parametrization is introduced and referred to later on but never really explained – I was a bit lost here.
>
> Great question that we unfortunately lack the space to answer, please ask again
>
> > Is the max-margin solution from theorem 1 actually used somewhere? The authors seem to only rely on the heuristic construction for the ReLU MLP introduced afterward.
>
> The max-margin solution from Theorem 1 offers a strong argument that the MLP discovers parallel/antiparallel weights. In this way, it justifies our hand-crafted solution presented in Section 3.1, and suggests that the MLP may indeed implement a weight configuration that resembles it. We have more to say, so please ask a follow-up if needed

---

> > ### Comment · Reviewer_rMwz · 2025-04-18
> >
> > _(This is TPC providing a pointer to the [**Official Comment**](https://openreview.net/forum?id=QaDfpS2dg2&noteId=cVHQ2dnRyl) from Reviewer rMwz starting with:_
> >
> > > Thanks for these comments. A few follow-up  ...
> >
> > _With this **Rebuttal Comment** posted on behalf of the Reviewer, the Authors can respond with one final **Reply Rebuttal Comment** during the Author-Reviewer Discussion phase.)_

---

> > > ### Author Response · Authors · 2025-04-18
> > >
> > > Thanks for the follow-up questions.
> > >
> > > > Do all converged models reach 100% accuracy on the training set?
> > >
> > > Most models do indeed converge to perfect or near-perfect accuracy on the training set. The exception are some lazy models trained on especially complex tasks (like the CIFAR-100 same-different task), which plateau at somewhat lower accuracy.
> > >
> > > > How exactly is test accuracy computed?
> > >
> > > Test accuracy is computed from taking the classification assigned the highest probability by the model. Your suggestion to average output probabilities is an interesting alternative, though not one we have considered explicitly. We imagine our qualitative results should still hold true: rich models should continue outperforming lazy models, since their higher test accuracy implies that they also assign higher probabilities to the correct classification.
> > >
> > > > In general, to update my assessment, it would help to point me towards concrete changes in the paper (e.g. by highlighting changes in the updated PDF and stating the corresponding page/paragraph) instead of doing a back and forth. Thanks in advance.
> > >
> > > Unfortunately, it appears we are unable to modify our PDF upload. Apologies for the inconvenience! The changes that most relate to your questions are the new Appendix F, which reviews rich/lazy parameterization as it applies to our setting, and the expanded Appendix G, which contains further details about the training and task setup.

---

### Official Review · Reviewer_ZYcH · 2025-03-28
**Solid paper with good evaluations and evidence**

**Soundness:** 3
**Clarity:** 2

**Comments:**

**Summary**
* The paper investigates the problem of learning whether two inputs (items) are equal under different representation learning regimes in artificial neural networks.

**Strengths**
* The paper has a solid theory, backed by recent papers in the literature (which they reference), but apply it to a new type of cognitive ability (equality reasoning). Their hypotheses are backed by well-formulated experiments.

**Limitations**
* The paper is limited in scope, since ``equality reasoning'' seems as a potentially narrow cognitive process.
* Though one experiment does not agree with the rest (CIFAR), it provides interesting avenues for future work/improvements.

1. Interest
* This will be of interest to researchers in representation learning (AI), as well some some subfields of computational neuroscience interested in applying neural networks to psychological tasks/neural data. I think the topic is a good fit for CCN.

2. Soundness
* The paper details both theoretical results, a long with an interpretable hand-crafted MLP solution to solve their task (equality learning).
* The paper also provides a number of experiments with MLPs that, for the most part, corroborate their theory. The one experiment that was not entirely consistent with their theory/hypothesis is the CIFAR experiment. As this is the most realistic experiment, I think it will be highly interesting to pursue this line of work in future iterations of this work.
* But overall, the results are convincing, given the combination of theory and experiments.

3. Clarity
* Overall, I found the clarity of the paper to be good. The writing was clear, and the figures appropriate. It may be helpful in a revision to provide a schematic depiction (visual of the hand-crafted neural network) as an additional figure to complement the mathematical description of the model described in section 3.1

Areas for improvement
* A visualization of the hand-crafted neural network solution presented in section 3.1
* A more in-depth understanding of why rich models perform poorly on the CIFAR task. What's most troubling (or interesting) is that it ends up performing the worst out of all models.

**Expertise:**

3

**Interest:**

2

---

> ### Author Rebuttal · Authors · 2025-04-14
>
> Thanks for the comments and suggestions! We’re glad you found our results to be clear and convincing. We’ve incorporated all edits in a revised manuscript uploaded to OpenReview. Responding to your specific suggestions:
>
> > The paper is limited in scope, since ``equality reasoning'' seems as a potentially narrow cognitive process.
>
> While equality reasoning is indeed a simple subset of general abstract reasoning, it remains a ubiquitous, representative, and extensively-studied behavior. Its simplicity allows us to use it as a tractable starting point for understanding much deeper questions about reasoning and generalization in neural networks.
>
>
> > A visualization of the hand-crafted neural network solution presented in section 3.1
>
> Great suggestion, we have added an illustration of the hand-crafted solution as Figure 5.
>
> > A more in-depth understanding of why rich models perform poorly on the CIFAR task. What's most troubling (or interesting) is that it ends up performing the worst out of all models.
>
> Regarding these results, while compiling this rebuttal, we discovered a small, unfortunate bug in our implementation that distorted the scaling across some experiments. We have updated all affected plots in the new manuscript uploaded to OpenReview. All theoretical predictions and qualitative trends continue to hold, though exact numerics differ somewhat. Apologies for this change!
>
> One result of this fix is that our CIFAR-100 plots are now more in line with our expectations. The richest model now performs competitively, whereas previously we now realize its performance was unstable due to the bug. However, the richest model does not always perform the best, and some contribution from overfitting likely still plays a role. With greater learning richness, we hypothesize that the model may be more susceptible to learning features specific to idiosyncrasies in the CIFAR-100 training data that generalize poorly, analogous to overfitting effects in classical statistics that degrade the performance of powerful models. We remain unsure what exactly about this dataset leads to non-monotonic improvement with learning richness, but anticipate studying this phenomenon more closely in future work.

---

> > ### Comment · Reviewer_ZYcH · 2025-04-18
> >
> > _(This is TPC providing a copy of the [**Official Comment**](https://openreview.net/forum?id=QaDfpS2dg2&noteId=wR2JfBdVIu) from Reviewer ZYcH with:_
> >
> > > I thank the authors for their clear rebuttal. I appreciate that they have identified a bug in their cifar analysis, and revised their results accordingly. Thank you for figure 5 -- it's a great visualization! The authors have addressed all my concerns, and I recommend acceptance.
> >
> > _With this **Rebuttal Comment** posted on behalf of the Reviewer, the Authors can respond with one final **Reply Rebuttal Comment** during the Author-Reviewer Discussion phase.)_

---

### Official Review · Reviewer_rize · 2025-03-30
**An exploration of how learning richness modulates equality reasoning in neural networks, with implications for both computational neuroscience and machine learning**

**Soundness:** 2
**Clarity:** 2

**Comments:**

### Summary of Claims and Approach:
- The manuscript claims that the degree of learning richness (i.e., the extent to which a network's internal representations adapt during training) is a key modulator of a network's ability to perform equality reasoning.
- It argues that networks in a rich learning regime develop conceptual representations that enable them to generalize from very few training examples and remain robust to changes in input dimensionality.
- Conversely, networks in a lazy regime, which minimally update their representations, rely on bruteforce memorization and are sensitive to perceptual variations.
- The approach combines theoretical analysis with simulations on synthetic and visual tasks (e.g., PSVRT, Pentomino, CIFAR-100).

### Strengths:
- Linking learning richness to the emergence of abstract reasoning offers a novel perspective on both neural network behavior and biological cognition.
- The simulations across various tasks demonstrate the predicted differences between rich and lazy regimes, reinforcing the theoretical claims.

### Limitations:
- While the concept of learning richness is well-explained, additional clarity on how hyperparameters (especially γ) are chosen and their interaction with other model parameters would help in replicating the results.

### Questions and Suggestions for the Authors:
- It would strengthen the work to include comparisons with other network architectures (e.g., CNNs or Transformers). While I am not suggesting this for the current study and understand the rationale for using MLP, I recommend comparing it with other architectures in future work.
- How might these findings inform the design of neural networks for real-world tasks that require abstract reasoning? Expanding on the potential applications could broaden the paper's impact.

**Expertise:**

1

**Interest:**

1

---

> ### Author Rebuttal · Authors · 2025-04-14
>
> Thanks for the comments! We’re glad you found the results to be novel and compelling. We’ve incorporated all edits in a revised manuscript uploaded to OpenReview. Responding to your specific questions:
>
> > While the concept of learning richness is well-explained, additional clarity on how hyperparameters (especially γ) are chosen and their interaction with other model parameters would help in replicating the results.
>
> The rationale behind our choice of $\gamma$ is detailed in Section 2. An additional overview on rich/lazy regime parametrization has been added as Appendix F. A full review of rich-regime parameterization is beyond our scope, but we cite other great resources for learning more like [Pehlevan and Bordelon 2023](https://mlschool.princeton.edu/sites/g/files/toruqf5946/files/documents/Princeton___Lecture_Notes_0.pdf). For full reproduction of our results, ready-to-run code in our GitHub repository is linked in Appendix G (but currently anonymized during review).
>
> > It would strengthen the work to include comparisons with other network architectures (e.g., CNNs or Transformers). While I am not suggesting this for the current study and understand the rationale for using MLP, I recommend comparing it with other architectures in future work.
>
> Thanks for the suggestion. We are indeed studying other architectures like Transformers, and look forward to presenting these results in future work.
>
> > How might these findings inform the design of neural networks for real-world tasks that require abstract reasoning? Expanding on the potential applications could broaden the paper's impact.
>
> Our primary finding is that learning richness offers a crucial hyperparameter that strongly influences a model’s success on reasoning tasks. As we explain further in the Discussion, learning richness is an oft-neglected hyperparameter that dramatically impacts a model’s performance. Particularly considering the surge of interest in LLMs applied to reasoning tasks, appropriate richness parameterization is vital. We expanded our discussion on this point.

---

> > ### Comment · Reviewer_rize · 2025-04-18
> >
> > _(This is TPC providing a copy of the [**Official Comment**](https://openreview.net/forum?id=QaDfpS2dg2&noteId=KyVpiRalVl) from Reviewer rize with:_
> >
> > > Their response adequately addresses my comments.
> >
> > _With this **Rebuttal Comment** posted on behalf of the Reviewer, the Authors can respond with one final **Reply Rebuttal Comment** during the Author-Reviewer Discussion phase.)_

---

### Official Review · Reviewer_yUiY · 2025-03-31
**An interesting analysis of equality reasoning that could be clearer about its scope**

**Soundness:** 3
**Clarity:** 3

**Comments:**

Overall, I really liked this manuscript and would recommend acceptance. It analyzes a topic that is broadly relevant for the cognitive computational neuroscience audience, is well written, and supports its claims through several theoretical analyses. That said, I think the authors should be clearer about the scope of their results and could do a better job of connecting these results to its implications for neuroscience. Below I will explain my ratings for interest, soundness, and clarity, and expand on this criticism.

**Interest**

This manuscript discusses equality reasoning, a topic that has received broad attention across neuroscience, cognitive science, and AI in recent years. Identifying sameness is a foundational ability in relational and symbolic reasoning and the conditions under which artificial neural networks can generalize on such tasks are important to clarify. Likewise, I believe that the theoretical framework that the authors provide will be interesting for cognitive scientists and neuroscientists, in particular making predictions for how generalization may scale with the number of symbols and what potential neural representations implementing equality reasoning may look like.

However, I think the authors should clarify the connection between their analysis and the ways in which same/different tasks have been studied in humans and animals. I think drawing an analogy between humans/animals and the lazy/rich spectrum in terms of the amounts of training data required makes sense, but I am more skeptical about the analogy in terms of sensitivity to spurious perceptual details. As far as I understand, in neuroscience experiments on the same/different task, perceptual solutions often refer to animals using an overall representation of coherence in a given stimulus to solve a same/different task (as stimuli with multiple identical stimuli will be more regular than stimuli with distinct stimuli) and one way in which experimenters have often probed this is by varying the number of stimuli that are either identical or different (as this should not affect a conceptual judgment of sameness, but stimuli with more objects would be easier to classify based on coherence). I think this is quite different from the stimulus dimensions $d$ the authors use here. I think being more specific in the manuscript about how the sensitivity to stimulus dimension relates to sensitivities in animal experiments would be important, both to substantiate the authors' claims and to clarify potential limitations.

Further, I think the manuscript would benefit from a more detailed discussion of potential behavioral and neural predictions the authors' theory would make (e.g. the predictions I point out above). What experiments could test these predictions? Is there existing data that could test some of these predictions? I think being more specific about these questions would be really useful, in particular to make these paper even more interesting to a neuroscience/cognitive science audience.

Finally, I wanted to point out that the methodology put forward in the paper may also be of interdisciplinary interest. Understanding when behavior/representations in neural network and humans or animals converge or are distinct is an important endeavor across neuroscience, cognitive science, and AI, and developing rigorous theoretical and empirical frameworks is useful for that endeavor. In particular, I strongly agree with the last sentence of the authors' discussion.

**Soundness**

The authors analyze the lazy and rich regime through several theoretical avenues and further provide empirical support for their theory, both in cases where their assumptions hold exactly and in cases where these assumptions are violated. Overall, this convincingly demonstrates that neural networks in the rich regime are able to generalize on the tasks they study.

One minor point: The authors only provide an upper bound on the test error of an MLP in the lazy regime, meaning that the theory does not establish the degree to which lazy-regime MLPs are *unable* to generalize successfully. They provide empirical evidence for this inability, however, so while this limits the scope of their theoretical results slightly, it does not affect the overall soundness of their results.

**Clarity**

I thought the paper was well written. I think the authors do a good job explaining why they leverage different theoretical perspectives. As noted above, I think the author could be clearer about the relevance of their paradigm to neuroscience.

In addition, I think it would be good to provide a bit more discussion on how previous studies have theoretically investigated relational reasoning and equality reasoning in particular. For example, Boix-Adserà et al. (2024, https://openreview.net/pdf?id=STUGfUz8ob) also study the same/different task in neural networks. Notably, they find that MLPs trained through gradient descent are worse at symbolic generalization on this task than Transformers. It would be useful to discuss these findings in relation to the findings presented in this paper.

Related to the point above, I think the authors could further elaborate on why the rich-regime networks can generalize from very few symbols even for large dimensionality. I was surprised at first by this finding (especially in the context of the paper cited above). One thing I think is relevant to point out is that, as far as I understand, the networks would be unable to generalize to any test symbol that is orthogonal to all training symbols or indeed the specific parallel/anti-parallel weights (which I think explains the relationship between this paper and Boix-Adserà et al.). However, the set of inputs that are exactly orthogonal have measure zero and so this doesn't affect accuracy. Since this paper refers, in part, to symbolic representations, and we might expect symbolic representations to generalize to arbitrary symbols, I think it would be important to clarify this point.

**Minor comments**
- L. 612: "sensitivity" -> "sensitive"
- L. 1068: Should this be $w_i=(w_i^1;w_i^2)$?
- L. 581ff.: What does it mean to say the rich model overfits? Do you see corresponding differences in the weight alignment for smaller gamma?
- I thought the discussion on the relevance of the rich regime for understanding human generalization in l. 616-626 was good. As a minor suggestion, I think the speculation in connecting this to synaptic plasticity specifically (l. 626-637) is a bit strenuous; at this point, we don't really have a framework for how we would want to connect rich-regime behavior to synaptic changes in the brain and I personally don't think it makes sense to even think about plasticity as the one-dimensional spectrum implied by "an unusual degree". The authors are careful in these speculations, so I'm fine with keeping this part in, but in my personal opinion, it doesn't really add anything to the previous part.

**Expertise:**

3

**Interest:**

3

---

> ### Author Rebuttal · Authors · 2025-04-14
>
> Thanks for the kind comments and suggestions! We’re glad you found our results to be strong and broadly appealing. We’ve incorporated all edits in a revised manuscript uploaded to OpenReview
>
> Unfortunately, due to the tight word limit imposed on rebuttals, we are unable to respond to all your questions. We offer brief answers below where possible, but please do ask additional questions to offer us the opportunity to elaborate. Apologies for the inconvenience!
>
> > Authors should clarify the connection... quite different from the stimulus dimensions d the authors use here
>
> Great suggestions. About perceptual behavior, certainly the number of stimuli is a perceptual feature that is commonly varied in experiments studying same-different tasks, as performed by Ed Wasserman and colleagues. Other perceptual variations they attempted include varying spatial organization and stimulus orientation. Under this framework, perceptual behavior does not necessarily imply a reliance on representational coherence. Rather, it suggests a broader sensitivity to perceptual features that are irrelevant to sameness. In our case, input dimension $d$ is a perceptual attribute that does not impact the final classification, making it a (spurious) perceptual detail.
>
> We have more to say, please ask a follow-up
>
> > What experiments could test these predictions?
>
> Great questions. We agree there are exciting predictions to derive from these results, and are in the early stages of considering experimental validation. Broadly, our results suggest that learning richness contributes to the observed spectrum of generalization behavior in animals on same-different tasks. Subjects that generalize quicker and more robustly may exhibit greater richness.
>
> We have more to say, please ask a follow-up
>
> > ...previous studies have theoretically investigated relational reasoning. For example, Boix-Adserà et al...
>
> Great question that we unfortunately lack the space to answer, please ask again
>
> > What does it mean to say the rich model overfits? Do you see corresponding differences in the weight alignment for smaller gamma?
>
> Another great question that we unfortunately lack the space to answer fully, please do ask again! Part of the issue was a small, unfortunate bug we discovered while compiling the rebuttal. Please check the updated manuscript for the corrected plot.
>
> > synaptic plasticity
>
> Thanks for the notes. We agree with your suggestion and have removed our sentences on synaptic plasticity.

---

### Meta-Review · Area_Chair_SJTp · 2025-04-29

**Ccn Recommendation:** Accept as Proceedings

**Metareview:**

I believe that the reviewers concerns have been adequately addressed (as several have acknowledged). I find this to be a clear and intriguing paper of broad interest to multiple areas of the field.

**Summary:**

This paper studies when MLPs generalize same/equality relations from the perspective of lazy vs. rich feature learning. Via theoretical and empirical results, the paper shows that the learning regime modulates the generalization performance. The reviewers generally agree that the work was sound and relatively clear. Given the historic and ongoing importance of relational reasoning (and the capacity of neural networks to acquire it) as a topic of debate in cognitive science, and the sustained interest in understanding the interaction between feature learning and generalization from a machine learning perspective, this work seems to me to be a strong submission that clearly has points of interest for several subsets of the CCN community.

**Expertise:**

3